# Aerosol delivery of SARS-CoV-2 human monoclonal antibodies in macaques limits viral replication and lung pathology

Daniel N. Streblow[1], Alec J. Hirsch [1], Jeffrey J. Stanton[2], Anne D. Lewis [2], Lois Colgin[2], Ann J. Hessell[2], Craig N. Kreklywich[1], Jessica L. Smith[1], William F. Sutton[2], David Chauvin[3], Jennifer Woo [3], Benjamin N. Bimber [2], Cierra N. LeBlanc[4], Sonia N. Acharya[4], Brian J. O'Roak [4], Harjinder Sardar[5], Mohammad M. Sajadi[6], Zahra R. Tehrani[7], Mark R. Walter [8], Luis Martinez-Sobrido [9], James J. Kobie [10], Rachel J. Reader [11], Katherine J. Olstad[11], Theodore R. Hobbs [2], Erica Ollmann Saphire [12], Sharon L. Schendel [12], Robert H. Carnahan[13], Jonas Knoch[14], Luis M. Branco[15], James E. Crowe Jr.[13], Koen K. A. Van Rompay [11], Phillip Lovalenti[3], Vu Truong[3] ✉, Donald N. Forthal[16] ✉ & Nancy L. Haigwood [2] ✉

Passively administered monoclonal antibodies (mAbs) given before or after viral infection can prevent or blunt disease. Here, we examine the efficacy of aerosol mAb delivery to prevent infection and disease in rhesus macaques inoculated with the severe acute respiratory syndrome coronavirus 2 (SARS-CoV-2) Delta variant via intranasal and intratracheal routes. SARS-CoV-2 human mAbs or a human mAb directed to respiratory syncytial virus (RSV) are nebulized and delivered using positive airflow via facemask to sedated macaques pre- and post-infection. Nebulized human mAbs are detectable in nasal, oropharyngeal, and bronchoalveolar lavage (BAL) samples. SARS-CoV-2 mAb treatment significantly reduces levels of SARS-CoV-2 viral RNA and infectious virus in the upper and lower respiratory tracts relative to controls. Reductions in lung and BAL virus levels correspond to reduced BAL inflammatory cytokines and lung pathology. Aerosolized antibody therapy for SARS-CoV-2 could be effective for reducing viral burden and limiting disease severity.

The global pandemic of severe acute respiratory syndrome coronavirus 2 (SARS-CoV-2) infection has led to over 6.6 million COVID-19-related deaths worldwide (WHO Coronavirus (COVID-19) Dashboard; covid19.who.int). SARS-CoV-2 is a betacoronavirus with high genetic similarity to coronaviruses in bats and pangolins[1,2]. Viral transmission is primarily through the respiratory route, and the epithelial cells of the upper and lower airways are the principal targets of infection. The virus binds to host cells via an interaction between the receptor-binding domain (RBD) of the viral spike protein (S) and angiotensin-converting enzyme 2 (ACE2)[3]. Cleavage of S into S1 and S2 domains is accomplished by furin in infected cells and further cleavage of S2 at the target cell surface by transmembrane serine protease 2 (TMPRSS2) leads to S2 domain-mediated viral fusion that promotes host cell entry[4]. SARS-CoV-2 dysregulates type I interferon (IFN), inflammatory, and T cell-mediated responses and stimulates a subsequent cytokine storm, leading to lung damage and lymphopenia[5–7].

Several animal models of SARS-CoV-2 infection have been developed that have provided critical data on pathogenesis, correlates of

protection, and efficacy of prophylactic and therapeutic interventions[8,9]. Experimental infection of nonhuman primates (NHPs), including rhesus macaques (*Macaca mulatta*), cynomolgus macaques (*Macaca fascicularis*), and pigtail macaques (*Macaca nemestrina*) with an initially described strain of SARS-CoV-2 (WA-1) did not result in severe disease but was characterized by rapidly peaking, high levels of viral replication for several days, viral shedding for ~2 weeks, and mild to moderate pulmonary disease with signs of interstitial pneumonia and accumulation of inflammatory monocytes, macrophages and neutrophils in the lungs[10], similar to that seen in humans[11]. In addition, NHPs develop immune responses to the virus comparable to those responses observed in infected humans. Thus, macaques are considered excellent models for testing vaccines, antivirals, or monoclonal antibody (mAb) immunotherapeutics targeting the virus as well as host-directed therapeutics to mitigate inflammation and pulmonary disease[8].

Despite the development of highly effective prophylactic vaccines based on the SARS-CoV-2 S protein, effective management of the pandemic is threatened by poor vaccine uptake or access, waning vaccine immunity, and the highly infectious nature of the virus, leading to continuously emerging SARS-CoV-2 variants[12–14]. Moreover, a vaccine may have to be given on multiple occasions and, importantly, may not provide adequate protection to the elderly (>65 yrs), the young (<2 yrs), or immunocompromised patient populations. Unfortunately, as many as one-fifth of individuals who recover from acute infection are plagued by unresolved symptoms for months, or long COVID syndrome. In addition, some evidence indicates that each re-infection is associated with increased cumulative risk of disease or death[15]. Continuous cycles of infection or re-infection, even when mild, keep the virus in circulation, meaning a slower end to the pandemic.

Drug therapy with nirmatrelvir/ritonavir blunts disease when given early following initial infection, and its use may help to prevent recurrence of severe disease and long COVID[16]. A recent study of the Janus kinase (JAK) inhibitor baracitinib in nonhuman primates infected with SARS-CoV-2 effectively reduced inflammation by eliminating the influx of non-alveolar macrophages into the lung, but this treatment did not reduce viral RNA[17]. As an alternative to small-molecule drugs, a number of mAbs directed against S protein, which is the target of neutralizing antibodies, have been shown to block infection in small-animal models when delivered prophylactically[18,19], and they can reduce viral RNA, infectious virus, and lung pathology in the post-exposure setting in these models[20]. When delivered at high doses by the intravenous route, several different combinations of these antibodies have shown promise in rhesus macaque models of COVID-19 to prevent infection[21,22] or to ameliorate disease when given after infection[23]. Several mAbs are currently in commercial use through the Emergency Use Authorization (EUA) program[24]. Passive antibody treatments using combinations of anti-SARS-CoV-2 mAbs can affect disease severity in humans, but they must be capable of neutralizing the infecting strain, and they are most effective when used early after exposure[25]. Furthermore, most authorized mAbs are delivered systemically by the intravenous or intramuscular route in a clinical setting, adding to inconvenience and medical costs. When delivered as immediate post-exposure prophylaxis for individuals who are virus positive but asymptomatic, mAb therapy can reduce disease severity and duration and thereby limit transmission[26]. To be maximally effective, such treatments need to be rapidly delivered to the tissues that are the first sites of viral replication and should be easy to administer, preferably in an outpatient or home setting.

Delivery of mAbs directly to airways via inhalation or intranasal application offers an alternative to systemically delivered mAbs for treating respiratory infections that is potentially more effective, more convenient, and potentially less costly than intravenous administration brought about from the dose-sparing effect of direct delivery to the target lung tissue. Apart from its use in RSV, inhaled or intranasal immunoglobulin has not been used to prevent infections in humans for decades. Recent studies in small animals[27,28] as well as in NHPs[29] suggest that antibodies delivered in this manner may be more effective than systemically administered antibodies in treating or preventing respiratory infections[30–32]. We evaluated the inhaled delivery of a combination of two human IgG mAbs with anti-SARS-CoV-2 neutralizing activity for their ability to reduce SARS-CoV-2 infection in rhesus macaques. Our findings demonstrate that inhaled anti-SARS-CoV-2 antibodies can decrease lung tissue virus load, inflammation, and disease even when delivered after infection.

## Results

### Delivery and uptake of nebulized mAbs in the respiratory tract of uninfected macaques

Prior to any in vivo work, each mAb in this study was characterized for specificity by antigen-specific ELISA and for neutralizing activity in SARS-CoV-2 focus-forming assays. Antibodies were delivered to sedated, recumbent animals via an aerosol mask, using an Investigational eFlow® Nebulizer System (PARI Respiratory Equipment Inc., Midlothian, VA). Twenty-two Indian-origin rhesus macaques were assigned for the entire study. To evaluate delivery and uptake, we tested four mAbs cloned from human subjects including three directed to the SARS-CoV-2 S-protein (CoVIC-96[33]; AR-701, a cocktail of mAbs AR-703[27,28] and AR-720 [also known as mAb 1213H7, see Piepenbrink et al.[28]]) and one antibody directed to the RSV fusion protein, which was used as a control[34]. The mAbs were previously characterized for in vitro binding and antiviral activity and they were expressed as IgG1s and purified for use in this study. To optimize the aerosol delivery methods for mAbs, two uninfected (SARS-CoV-2-naïve) Indian-origin rhesus macaques were given a single dose of nebulized CoVIC-96 at 10 mg/kg. At 6, 24, 48, 72 and 96 h post delivery, nasopharyngeal (NAS) and oropharyngeal (OP) swabs, blood plasma, and bronchoalveolar lavage (BAL) samples were obtained for quantification of antibody concentrations by antigen-specific ELISA. mAb sampling from the NAS and OP swabs and BAL fluid at various time points after dosing captured just a small region of each tissue, thus providing only qualitative estimates of relative regional concentrations. The mAb was detected in NAS, OP, and BAL samples immediately following administration, and the levels diminished more quickly in the upper airway compartment (NAS and OP samples), whereas the antibody was readily detected at the 96 h timepoint in BAL samples (Supplementary Fig. 1). Following clearance of this mAb, these two NHPs were given the 2-antibody cocktail AR-701 (stem helix S2 domain binding, $NT_{50}$ 1,540 ng/mL against the Delta variant[27,28]) and AR-720 mAb (RBD S1 domain binding, $NT_{50}$ 53 ng/mL against the Delta variant[27,28]) at 37.5 mg total mAbs/kg (nebulizer device-loaded dose or device dose), and the NAS, OP, plasma, and BAL samples were obtained longitudinally. Like COVIC-96, these mAbs were readily detected in all samples with modest animal-to-animal variation (Supplementary Fig. 1). Antibody concentrations diminished rapidly over the 96 h observation time, with the longest persistence in BAL samples and the most rapid decline in NAS and OP samples. Concentrations measured by quantitative ELISA were verified by gel electrophoresis (Supplementary Fig. 1). As expected, the amount of mAbs detected at the peak immediately after nebulization was higher in the animals receiving the AR-701 cocktail, which were dosed ~3.7-fold higher concentrations of antibodies as compared to the CoVIC-96 antibody. No adverse events were reported for either of the animals and they were returned to the colony for other studies. Based on results from these studies, a final dose of 60 mg (~15 mg/kg) by the inhalation route was chosen for the SARS-CoV-2 challenge study.

### Delivery and uptake of nebulized mAbs in SARS-CoV-2-challenged macaques

The study design for the 20 SARS-CoV-2-inoculated animals is outlined in Fig. 1a. Animals were distributed randomly into five cohorts (*n* = 4

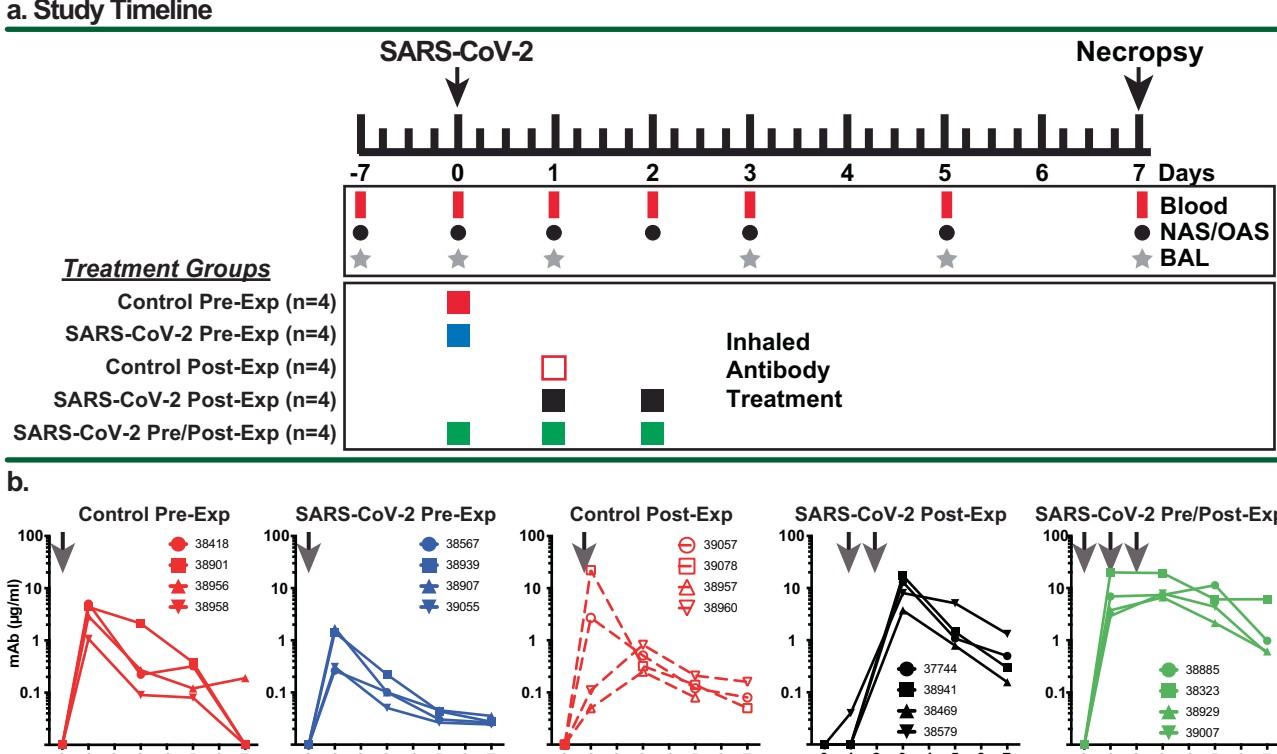

**Fig. 1 | Summary of study design and detection of inhaled aerosolized mAbs in BAL samples. a** Study design and timeline for experiments involving 5 treatment groups of rhesus macaques (*n* = 4 per group). The animals were infected with the SARS-CoV-2 Delta strain by the intranasal and intratracheal routes. Blood, BAL and swab samples were collected at the indicated times, and the animals were euthanized for necropsy at 7 dpi. The Control and SARS-CoV-2 Pre-Exposure Groups each received a single administration of 60 mg (~15 mg/kg) nebulized antibody (anti-RSV control or anti-SARS-CoV-2 mAb cocktail, respectively) at approximately 5.8 mg/kg of inhaled dose at 6 h prior to challenge. The Control Post-Exposure group received a single administration of 60 mg aerosolized, inhaled anti-RSV control antibody at 24 h post-challenge dosed at approximately 5.8 mg/kg of inhaled dose. The SARS-CoV-2 Post-Exposure group received two administrations of 60 mg nebulized anti-

SARS-CoV-2 mAb cocktail dosed at 5.8 mg/kg at 24 and 48 h post challenge. SARS-CoV-2 Pre/Post-Exposure Group received three administrations of aerosolized, inhaled anti-SARS-CoV-2 mAb cocktail at 6 h pre challenge and at 24 and 48 h post challenge. **b** Qualitative measures of mAb in BAL were determined from clarified samples from each animal in the group collected at 0, 1, 3, 5, and 7 dpi using antigen-specific quantitative ELISA. Arrows indicate timing of aerosolized, inhaled mAb treatment for each group. Symbols refer to the four different animals in each group, and colors were assigned as follows: solid red = Control Pre-Exp group; solid blue = SARS-CoV-2 Pre-Exp group; open red = Control Post-Exp group; solid black = SARS-CoV-2 Post-Exp group; and solid green = SARS-CoV-2 Pre/Post-Exp group. Source data are provided as a Source Data file.

each) that were handled sequentially due to space limitations in ABSL-3 housing and study logistics. Treatment in the five cohorts differed by antibody and time of antibody administration with regard to viral challenge. Because aerosol administration of SARS-CoV-2 antibodies had not been tested previously in primates in the setting of viral infection, we designed this study to test antibody delivery prophylactically, as well as therapeutically. Since we knew that the nebulized human mAbs are cleared quickly from the lungs of rhesus macaques, we also tested two multidosing regimens to determine which of these therapies might improve outcomes. Baseline samples were collected from each of the animals 7-21 days prior to challenge. Two groups received a single 60 mg dose of a control antibody (anti-RSV 25P13 at ~15 mg/kg) at either 6 h pre-exposure (Control Pre-Exposure) or 24 h after exposure (Control Post-Exposure), respectively. Animals in the anti-SARS-CoV-2 mAb (AR-701)-treated cohorts also received one to three 60 mg aerosol doses, depending upon the dosing regimen group. One group received an aerosol dose 6 h prior to exposure (SARS-CoV-2 Pre-Exposure); one group was treated with two aerosol doses given 18 and 42 h after exposure (SARS-CoV-2 post exposure); and one group received three 60 mg aerosol doses of anti-SARS-CoV-2 mAbs 6 h prior to, 18 h after and 42 h after exposure (SARS-CoV-2 Pre/Post exposure). All challenges were with 10^6 pfu SARS-CoV-2 (Delta variant) via intranasal (i.n.) and 10^6 pfu via intratracheal (i.t.)

inoculation. NAS and OP surfaces were swabbed at 0, 1, 2, 3, 5, and 7 dpi, and the swab was resuspended in PBS for evaluation of infectious virus and viral RNA. The lower airway was sampled via BAL on days 1, 3, 5 and 7 after virus inoculation. Animals were euthanized on 7 dpi for histological analysis of the lungs and detection of viral RNA in tissue samples. Details of nebulization time and antibody concentrations are shown in Table 1.

We observed that a single dose of aerosolized SARS-CoV-2 or RSV mAbs resulted in a peak concentration between 1 and 10 μg/mL in BAL fluid that decreased but persisted in the lungs of all the treated animals, (Fig. 1b). Not surprisingly, repeated post-exposure dosing with the SARS-CoV-2 mAb combination resulted in 10-fold higher peaks and more persistent levels than single dose mAbs during the seven-day period. Dosing with AR-701 mAbs both pre- and post-exposure resulted in 10- to 50-fold higher levels of antibodies in the lung, in the range of 10 μg/mL, and these levels persisted for the duration of the experiment. We detected 10- to 100-fold higher concentrations of mAbs in NAS and OP swabs, but these mAbs were cleared quickly (>90% cleared within 6 h of dosing), while the initially low levels in the plasma increased gradually over time, indicating transfer of mAbs from the lung to systemic vasculature (Supplementary Fig. 2). The concentration and kinetics of the AR-701 SARS-CoV-2 mAbs were similar to that of the RSV mAb in all sample types, thus showing that the presence of

**Table 1 | Antibody concentrations and nebulization times**

| Treatment group | Animal ID | Nebulization date | Antibody amount (mg) | Antibody volume (mL) | Animal weight (kg) | Antibody concentration (mg/mL) | Nebulization Time |
|---|---|---|---|---|---|---|---|
| Control Pre-exposure | 38418 | 4/11/22 | 60 | 4.5 | 4.75 | 13.3 | 7 min 55 s |
| | 38901 | 4/11/22 | 60 | 4.5 | 4.70 | 13.3 | 8 min 33 s |
| | 38956 | 4/12/22 | 60 | 4.5 | 4.05 | 13.3 | Not recorded |
| | 38958 | 4/12/22 | 60 | 4.5 | 4.80 | 13.3 | Not recorded |
| SARS-CoV-2 Pre-exposure | 38567 | 4/20/22 | 60 | 2.5 | 5.25 | 24 | 6 min 20 s |
| | 38939 | 4/20/22 | 60 | 2.5 | 4.95 | 24 | 5 min 30 s |
| | 38907 | 4/21/22 | 60 | 2.5 | 5.50 | 24 | 5 min 6 s |
| | 39055 | 4/21/22 | 60 | 2.5 | 4.60 | 24 | 6 min 22 s |
| Control Post-exposure | 39057 | 4/29/22 | 60 | 4.5 | 3.65 | 13.3 | ~8 min |
| | 39078 | 4/29/22 | 60 | 4.5 | 4.25 | 13.3 | ~8 min |
| | 38957 | 4/30/22 | 60 | 4.5 | 4.85 | 13.3 | Not recorded |
| | 38960 | 4/30/22 | 60 | 4.5 | 4.10 | 13.3 | Not recorded |
| SARS-CoV-2 Post-exposure | 37744 | 6/2/22 | 60 | 3.0 | 4.90 | 20 | 9 min |
| | 38579 | 6/3/22 | 60 | 3.0 | 4.90 | 20 | 10 min |
| | 38469 | 6/2/22 | 60 | 3.0 | 4.15 | 20 | 10 min |
| | 38941 | 6/3/22 | 60 | 3.0 | 3.85 | 20 | 9 min |
| | | 6/3/22 | 60 | 3.0 | 3.70 | 20 | 6 min |
| | | 6/4/22 | 60 | 3.0 | 3.55 | 20 | 6 min |
| | | 6/3/22 | 60 | 3.0 | 3.90 | 20 | 6 min 12 s |
| | | 6/4/22 | 60 | 3.0 | 3.90 | 20 | 6 min 37 s |
| SARS-CoV-2 Pre/Post-exposure | 38323 | 5/16/22 | 60 | 3.0 | 5.05 | 20 | 5 min 10 s |
| | 38885 | 5/17/22 | 60 | 3.0 | 5.05 | 20 | 5 min 30 s |
| | 39007 | 5/18/22 | 60 | 3.0 | 4.95 | 20 | 5 min 35 s |
| | 38929 | 5/16/22 | 60 | 3.0 | 4.05 | 20 | 5 min 30 s |
| | | 5/17/22 | 60 | 3.0 | 3.80 | 20 | 5 min 33 s |
| | | 5/18/22 | 60 | 3.0 | 3.75 | 20 | 6 min 12 s |
| | | 5/17/22 | 60 | 3.0 | 4.25 | 20 | 8 min 30 s |
| | | 5/18/22 | 60 | 3.0 | 4.15 | 20 | 8 min 25 s |
| | | 5/19/22 | 60 | 3.0 | 4.15 | 20 | 10 min |
| | | 5/17/22 | 60 | 3.0 | 4.95 | 20 | 7 min 10 s |
| | | 5/18/22 | 60 | 3.0 | 4.80 | 20 | 7 min 55 s |
| | | 5/19/22 | 60 | 3.0 | 4.75 | 20 | 8 min 30 s |

All nebulizations were performed with 2 L/min oxygen flow rate. Liquid holdup was not measured but is estimated to be <0.5 mL.

infectious virus did not impact the kinetics of the passively delivered antibodies.

**Clinical outcomes**

A scoring system was used to monitor the animals throughout the study to determine if clinical outcomes would improve in treated animals compared to control animals (the scoring system was adapted from Van Rompay et al.[23]). As shown in Table 2, most animals had no or only mild clinical signs, revealing no differences between treatment and control animals. One animal in the control pre-exposure group and two animals in the anti-SARS-CoV-2 pre/post treatment group were observed to be coughing on days 5 and/or 6 post infection. Each of those animals received analgesia due to the potential for discomfort. One of the animals reported as coughing in the control pre-exposure group became dyspneic while recovering from BAL on day 5, and then improved clinically with brief oxygen therapy and analgesia. On day two, one animal in the control pre-exposure group was observed with nasal discharge that was not suspected to be due to a procedure. Minor weight loss is expected with frequent sedation of macaques as occurred in this study; however, no animal lost greater than 6% body weight from D0 to D7 and there was no apparent correlation between weight loss and treatment group (Table 1). The single and repeat antibody treatments were also well tolerated.

**Inhaled antibody treatment reduces viral load in lungs**

In BAL samples of the lower airway, we observed that viral genomic RNA peaked at 1 dpi, while titers of infectious virus generally peaked at 3 dpi in control animals (Fig. 2a, c). Notably, viral RNA was reduced by ~3 $\log_{10}$ at 1 dpi in animals that received pre-exposure anti-SARS-CoV-2 antibodies or both pre- and post- challenge antibodies and remained 1–2 $\log_{10}$ below controls on average (Fig. 2a). Infectious

virus was below the level of detection throughout the sampling period in SARS-Co-V-2 mAb pre-treated and pre-post treatment groups (Fig. 2c). In the post-infection treatment group, levels of viral RNA and infectious virus were similar to controls on 1 dpi (sampled prior to antibody administration), but infectious virus titers were below or at the level of detection following treatment and viral RNA was reduced ~1 $\log_{10}$ relative to control groups. The total burden of viral RNA and infectious virus present in BAL, NAS, and OP over the course of the experiment was calculated as area under the curve (AUC) for each animal and mean AUC calculated for each cohort, combining the two control groups into one (Fig. 2b, d). As shown, treatment with anti-SARS-CoV-2 pre/post-exposure resulted in significantly reduced viral replication in BAL, and NAS, and significantly lower infectious virus in BAL, NAS, and OP. Pre-exposure with anti-SARS-CoV-2 antibodies significantly reduced both viral replication and infectious virus in BAL, while viral load was reduced to a greater extent in the upper airway (NAS) by antibody given post-exposure. Levels of viral RNA were not significantly affected in OP swabs by treatment, and infectious titer was only reduced in OP swabs in animals treated with anti-SARS-CoV-2 pre- and post-exposure.

The upper airway, sampled by NAS and OP swabs, showed detectable infectious virus at 1 dpi, and detection throughout the sampling period for control animals that received anti-RSV mAb pre- or post-exposure (Fig. 2c). SARS-CoV-2 antibody treatment with a single 60 mg dose pre-exposure resulted in a 1-day delay in detection of infectious virus (NAS: at 3 dpi Control Pre-Exposure vs. SARS-CoV-2 Pre/Post Exposure, $P = 0.0146$; 5 dpi Control Pre-Exposure vs. SARS-CoV-2 Pre/Post-Virus Exposure, $P = 0.004$; Control Pre-Exposure vs. SARS-CoV-2 Post-Exposure, $P = 0.007$; and Control Post-Exposure vs. SARS-CoV-2 Pre/Post Exposure, $P = 0.0113$). Virus was detected at 1 dpi in animals treated with two 60 mg doses of SARS-CoV-2 mAbs post-

**Table 2 | Animal demographics and symptoms**

| Treatment group | Animal ID | Age | Sex | Weight loss (% of initial) | Symptom (day) | Treatment (day) |
|---|---|---|---|---|---|---|
| Control Pre-Exposure | 38418 | 2 yrs, 360 d | M | 2% | Cough (5,6) | Oxygen (5), Meloxicam (5,6) |
| | 38901 | 2 yrs, 348 d | M | 0% | Nasal discharge (2) | |
| | 38956 | 3 yrs, 3 d | M | 2% | | |
| | 38958 | 2 yrs, 347 d | M | 0% | | |
| SARS-CoV-2 Pre-Exposure | 38567 | 2 yrs, 330 d | M | 2% | | |
| | 38939 | 3 yrs, 24 d | M | 0% | | |
| | 38907 | 3 yrs, 11 d | M | 2% | | |
| | 39055 | 3 yrs, 5 d | M | 0% | | |
| Control Post-Exposure | 39057 | 2 yrs, 306 d | F | 6% | | |
| | 39078 | 2 yrs, 305 d | F | 5% | | |
| | 38957 | 3 yrs, 83 d | F | 0% | | |
| | 38960 | 3 yrs, 28 d | F | 3% | | |
| SARS-CoV-2 Post-Exposure | 37744 | 2 yrs, 347 d | F | 0% | Cough (6) | Meloxicam (6) |
| | 38941 | 3 yrs, 123 d | F | 5% | Cough (5,6) | Meloxicam (6) |
| | 38469 | 3 yrs, 36 d | F | 5% | | |
| | 38579 | 2 yrs, 342 d | F | 4% | | |
| SARS-CoV-2 Pre/Post- Exposure | 38885 | 2 yrs, 357 d | M | 1% | | |
| | 38323 | 3 yrs, 73 d | M | 0% | | |
| | 38929 | 3 yrs, 54 d | F | 1% | | |
| | 39007 | 2 yrs, 347 d | F | 5% | | |

exposure at levels similar to controls, followed by reduction in titers relative to controls at later sample times.

At 7 dpi, tissue sections were collected from individual lung lobes, trachea, and tracheobronchial lymph nodes. RNA was isolated from tissues and viral loads were quantified by qRT-PCR. Individual sections sampled are shown in Fig. 3a and Supplementary Fig. 3. Animals treated with the control RSV mAb showed high levels (>10$^5$ genomes/μg RNA) throughout the lungs. In contrast, substantially less viral RNA was detected in lung tissue from animals treated with AR-701. Notably, animals that received AR-701 post-exposure had high levels of viral RNA in the right caudal lobe, but lower levels in other regions, suggesting that our method of virus delivery favors deposition of the viral inoculum in the right caudal lobe and that antibody treatment inhibited further spread. Samples of the nasal turbinate and trachea showed no clear decrease in viral loads (Fig. 3b, c and Supplementary Fig. 3), consistent with the rapid clearance of antibodies from the upper airway. Similar to the trend of decreased levels of SARS-CoV-2 RNA detection in the lungs following mAb treatment, reduced levels of viral RNA were detected in both heart and spinal cord relative to the control mAb treatment groups (Fig. 3d, e). Deep sequencing revealed relatively few SARS-CoV-2 mutations in any of the tissue samples, with the majority of mutations being rare (<15% of the quasispecies). There were no significant differences in mutation rate between the cohorts, including within the spike gene (Supplementary Fig. 7b and Supplementary Tables 1–5). The only mutation detected in >15% of reads in multiple samples is a synonymous mutation (15953:C>T) also detected in the challenge stock. Overall, these data demonstrate that administration of AR-701 does not result in altered patterns of viral mutation or select for escape mutations in the spike protein in the timeframe examined in this study.

**Inhaled antibody treatment reduces lung pathology in infected animals**

All animals in the control groups exhibited variable degrees of bronchointerstitial pneumonia (Fig. 4a). The distribution within lobes was patchy to multifocally extensive, involving up to 80% of a given lung lobe in the most affected animals. The most severe lesions occurred in right and left caudal lung lobes and the right accessory lobe. Microscopically, alveolar walls were expanded by infiltrating lymphocytes, neutrophils, and macrophages, prominent type II pneumocyte hyperplasia and variable amounts of fibrin (Fig. 4a and Supplementary Fig. 5a, b). Alveolar lumina often contained high numbers of macrophages, neutrophils, fewer lymphocytes, protein-

rich edema fluid, and variable amounts of loose and compacted fibrin (Fig. 4a and Supplementary Fig. 5d). Bronchioles were often lined by attenuated epithelium that lacked cilia and were infiltrated by numerous neutrophils and lymphocytes (Supplementary Fig. 5e). Bronchi were more commonly lined by slightly hyperplastic epithelium infiltrated by numerous lymphocytes and neutrophils (Supplementary Fig. 5g). Syncytia were common in both alveolar and bronchiolar epithelium. Airways contained moderate numbers of neutrophils and macrophages admixed with small amounts of fibrin and mucin. In areas with abundant alveolar edema, fluid spilled into adjacent airways. Bronchus-associated lymphoid tissue was generally hyperplastic with prominent follicle development (Supplementary Fig. 5c). In larger pulmonary vessels, the endothelium was occasionally disrupted, and endothelial cells were rounded up, infrequently pyknotic, and interspersed with infiltrating leukocytes indicative of endotheli<br>tis (Supplementary Fig. 5f). The animal receiving control mAb that displayed clinical signs (Table 1) had extensive areas of severe, necrohemorrhagic pneumonia characterized by septal necrosis, alveolar hemorrhage, edema fluid, and abundant fibrin, as well as fibrinoid necrosis of small vessels (Fig. 4a and Supplementary Fig. 5h).

In contrast, the macaques in all of the SARS-CoV-2 mAb treatment groups exhibited minimal to only focally severe areas of bronchointerstitial pneumonia (Fig. 4a). This difference is reflected in the semiquantitative lung pathology scores that demonstrate significant differences between the control mAb treated animals and those receiving the SARS-CoV-2 mAbs, regardless of treatment regimen (Fig. 4d and Supplementary Table 6). Most histologic features noted in the control mAb groups were present in the SARS-CoV-2 mAb treatment groups, but to a much lesser extent (ranging from 10–35% vs. up to 80% in the control groups). In all animals in the SARS-CoV-2 mAb pre- and pre/post-treatment groups and one animal in the post-treatment group (38469), pulmonary inflammation was minimal to mild, widely scattered, and the type II pneumocyte hyperplasia was markedly less robust (Fig. 4a). Similar to the animals in the control groups, the caudal and accessory lung lobes were usually more affected than the other lobes. In general, animals with post-infection treated animals were protected from severe lung pathology, but three animals (37744, 38579, and 38941) exhibited scattered foci of moderate to severe inflammation and prominent type II pneumocyte hyperplasia (Fig. 4a). In two of these animals (37744 and 38579), aveolar protein-rich edema fluid was present. Necrohemorrhagic pneumonia with rare hyaline membranes was detected in one of these macaques (38579) (Fig. 4c). These microscopic findings were similar to those observed in

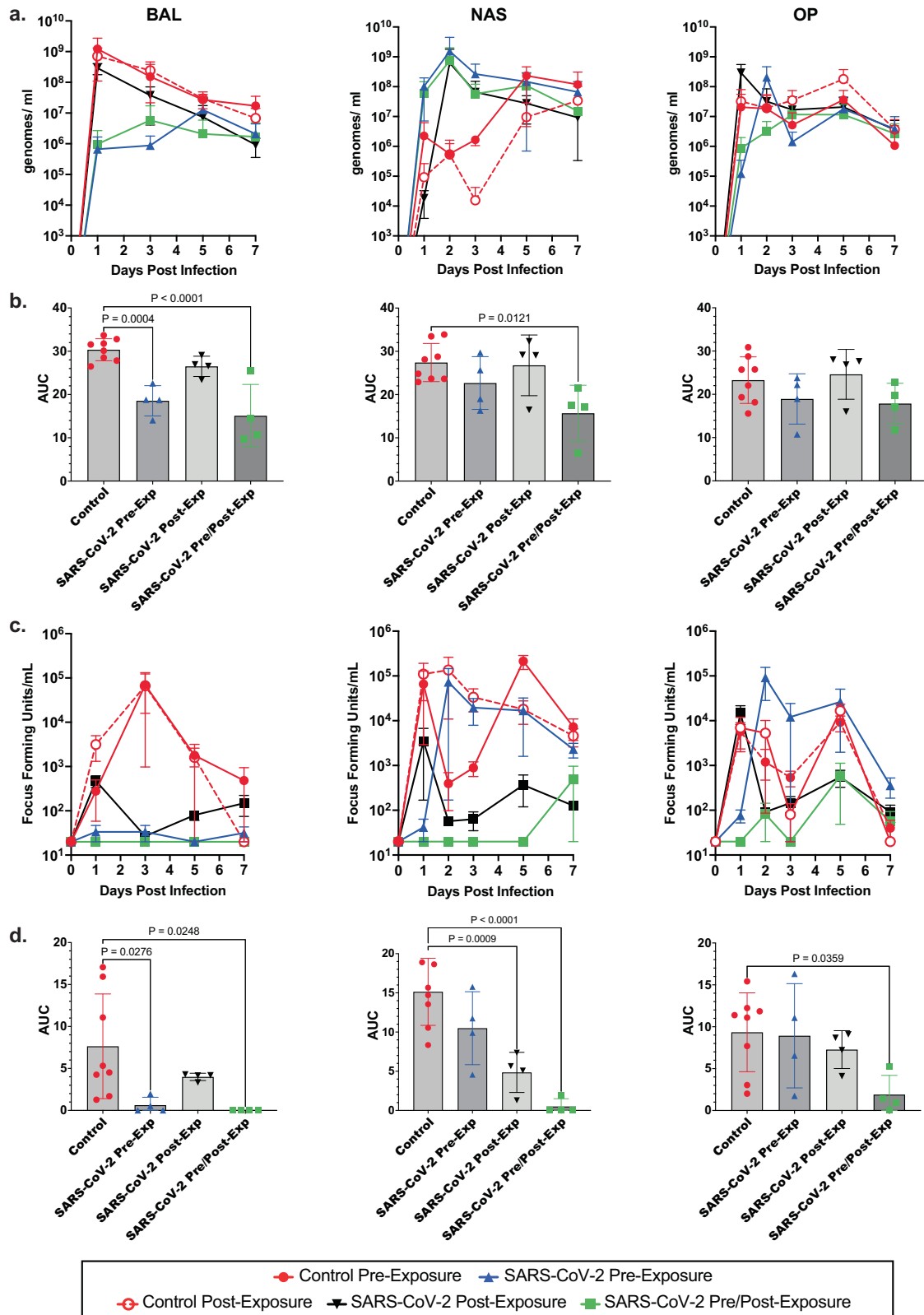

**Control Pre-Exposure** — ●
**Control Post-Exposure** — ○
**SARS-CoV-2 Pre-Exposure** — ▲
**SARS-CoV-2 Post-Exposure** — ▼
**SARS-CoV-2 Pre/Post-Exposure** — ■

macaque 38418 (Fig. 4b) in the control mAb pre-treatment group but very little of the lung was affected. Rare syncytia were present in alveolar and bronchiolar epithelium and were more common in the three animals with moderate to severe inflammation. Changes in the intrapulmonary conducting airways such as attenuated bronchiolar epithelium, inflammatory infiltrates and variable bronchial epithelial hyperplasia were present, but the degree and extent were less than in

the control groups. As in the control groups, hyperplasia of the bronchus-associated lymphoid tissue was often present. None of the macaques in the SARS-CoV-2 mAb treatment groups exhibited endothelialitis.

Findings in the nasal cavity, nasopharynx and trachea were similar in both control and treatment groups; the degree ranged from minimal to severe. The nasal and nasopharyngeal mucosa had variable

**Fig. 2 | Inhaled aerosolized anti-SARS-CoV-2 mAb treatment reduces SARS-CoV-2 in respiratory tract. a** SARS-CoV-2 viral RNA was quantified in BAL, NAS, and OP samples by qRT-PCR using a standard curve. Mean values ±Standard deviation (SD) for each cohort at each day post infection are shown. **b** Area under the curve (AUC) based on viral RNA levels was for determined each animal. Controls were considered as single cohort for this analysis. Individual values (symbols), mean AUC (bars), and SD (error bars) are shown. Statistical significance was determined by Tukey one-way ANOVA, comparing each group to controls. **c** Infectious virus in BAL, NAS, and OP was quantified by focus-forming assay. Mean values ± SD for each cohort at each day post infection are shown. **d** AUC (mean values ± SD) based on infectious titers was determined as above, and statistical significance was determined by Tukey one-way ANOVA. *N* = 4 biologically independent animals per group. Colors and symbols are: solid red circle, Control Pre-Exp group; solid blue triangle, SARS-CoV-2 Pre-Exp group; open red circle, Control Post-Exp group; solid black triangle, SARS-CoV-2 Post-Exp group; and solid green square, SARS-CoV-2 Pre/Post-Exp group. Exact *P* values are shown where greater than *P* = 0.05. Source data are provided as a Source Data file.

epithelial hyperplasia, loss of cilia, and infiltration by neutrophils, plasma cells and lymphocytes (Supplementary Fig. 6a). Mucosa-associated lymphoid tissue was often prominent. Rarely, the nasal mucosa was ulcerated and hemorrhagic (Supplementary Fig. 6b). In the trachea, there was segmental loss of cilia, epithelial attenuation and variable infiltration by neutrophils and lymphocytes (Supplementary Fig. 6c). The lamina propria and submucosa were edematous and contained moderate numbers of lymphocytes, macrophages, neutrophils and rare plasma cells. Multifocally, the tracheal mucosa was denuded and there was hemorrhage and vascular fibrinoid necrosis and fibrin thrombi in the subjacent tissue (Supplementary Fig. 6d). Scant luminal exudates were composed of neutrophils admixed with erythrocytes, mucin and fibrin.

### Altered cytokine expression in BAL of treated animals

We analyzed the expression of 37 cytokines in BAL samples from infected animals. In control animals, expression of several of these cytokines was observed to increase following infection (Fig. 5, red lines) in agreement with previously published studies[35,36]. For most cytokines, expression in treated animals was reduced compared to controls (e.g. IL-1b, IL-6, TNFα), in agreement with our observations of reduced immune cell infiltration. Notably, expression of a subset of cytokines was increased compared to controls in animals treated with the AR-701 cocktail given post exposure. (Fig. 5, black lines, e.g. CXCL8, CXCL9, CXCL10; and Supplementary Table 7). A two-way ANOVA statistical analysis was performed, and the following differences were significant: CXCL10 at 3 dpi control vs. SARS-CoV-2 pre/post exposure, *P* = 0.0117; CXCL11 at 3 dpi control post-exposure vs. SARS-CoV-2 pre/post-exposure, *P* = 0.015 and control post-exposure vs. SARS-CoV-2 post-exposure, *P* = 0.0172; IL1RA at 3 dpi control post-exposure vs. SARS-CoV-2 pre/post-exposure, *P* = 0.0198, and control post-exposure vs. SARS-CoV-2 post-exposure, *P* = 0.0316; IFNα at 1 dpi control post-exposure vs. SARS-CoV-2 pre/post-exposure, *P* = 0.0303.

## Discussion

This study is the first to show that human mAbs can be effective against early SARS-CoV-2 infection in primates when delivered by the aerosol route. Previous work had shown the feasibility of delivering IgG to the lung of rodents[27,28] and macaques[29] following nebulization. Our data show the efficacy of aerosolized immunoprophylaxis against SARS-CoV-2 infection, which targets primarily the respiratory tract and damages lung tissue. The rapid delivery of mAbs to the lung significantly reduced virus replication in BAL and subsequent pulmonary inflammation, both histologically and as measured by secreted cytokines.

In comparison to intravenous mAb infusions, aerosol application resulted in rapid mAb delivery to the lungs at concentrations 1–10 μg/mL at the peak, which was well above the mAb IC$_{50}$ (AR-701 mAb cocktail exhibited an IC$_{50}$ of ~53 ng/ml in vitro against the Delta variant). Moreover, we observed very significant viral clearance, based on viral RNA, sgRNA, and viral culture. The profound effect of mAbs was apparent in the first three days following virus exposure and most pronounced in animals that received mAbs pre-exposure. Data for these viral assays are internally consistent in showing that all three anti-SARS-CoV-2 mAb treatment regimens reduced or cleared infectious virus in BAL and viral RNA from lung tissues, with the greatest effect seen for the pre-and pre/post-exposure group. Levels of vRNA in other tissues tested were much lower than levels in the respiratory tract, consistent with the concept that the respiratory tract is a critical first target for therapies. The observation of no significant differences in viral RNA between control and treatment groups in tissues beyond the lungs is suggestive of inefficient transfer of antibodies from the nasal passages and lungs into the plasma during the timeframe of this study. This study was not designed to obtain pharmacokinetics of the antibody delivery or to determine the dose delivery efficiency of inhaled delivery as compared to systemic delivery but rather to observe clinical effects using a high nebulized antibody dose that is delivered at varying times relative to viral challenge and at different dosing frequency. We utilized 60 mg of mAbs, translating to a device-loaded dose of approximately 15 mg/kg, an estimated inhaled dose of 5.8 mg/kg, and a lung-deposited dose of 0.13 mg/g lung tissue. Repeated dosing was more effective in retaining mAb concentrations in the lung, as well as in reducing viral replication. The SARS-CoV-2 mAbs as well as the RSV mAb were engineered to extend their residence time in the plasma, and it is not known how these alterations may have affected turnover in the respiratory tract.

In a virus challenge study in which hamsters received an anti-SARS-CoV-2 mAb by inhalation lung-deposited dose levels as low as 0.001 mg/g of lung tissue showed significant reduction of lung viral burden[28], implying that a clinically efficacious dose by inhalation would be only 2 mg, which is 30-fold less than levels tested in our study. It is not known whether starting with a mAb dose lower than 60 mg in the nebulizer would provide similar efficacy in reducing viral burden in NHPs. However, the doses tested in hamsters were still very effective in reducing viral replication and thereby preventing lung damage, suggesting that aerosol delivery could be effective at lower doses than parenteral delivery requires. The anatomy of the primate respiratory tract is much more similar to humans than that of rodents, which may explain differences in viral replication as well as lung deposition of the mAbs. The current study thus provides a benchmark from which to expand such studies in primate models and in the clinic. We observed that delivery of mAb using a silicone facemask resulted in significant loss of antibodies through the egress port designed for $CO_2$ escape, particularly with the flow of oxygen to facilitate uptake. These are experimental variables that would not hold in clinical studies. Adults and children can be taught to use a nebulizer with significantly less loss of material, but due to the sedation required to safely perform this experiment, this was not the case for this study in macaques.

Lung pathology is the most sensitive parameter to demonstrate efficacy, and it is a surrogate marker for clinical disease following SARS-CoV-2 infection in NHP models. The differences in the degree of lung histopathology between treated and control groups were striking and these differences were reflected in the lung pathology scores. Although the lungs of treated and control animals had similar patterns of inflammation, the severity of inflammation and the extent of tissue involvement was markedly attenuated in treated animals. These differences corresponded well with both the reduced pulmonary viral loads, particularly of infectious virus, and the reduction of BAL-associated pro-inflammatory cytokines Il-1β, IL-6, IFNα, TNFα, and G-CSF compared to control animals. SARS-CoV-2 is hypothesized to

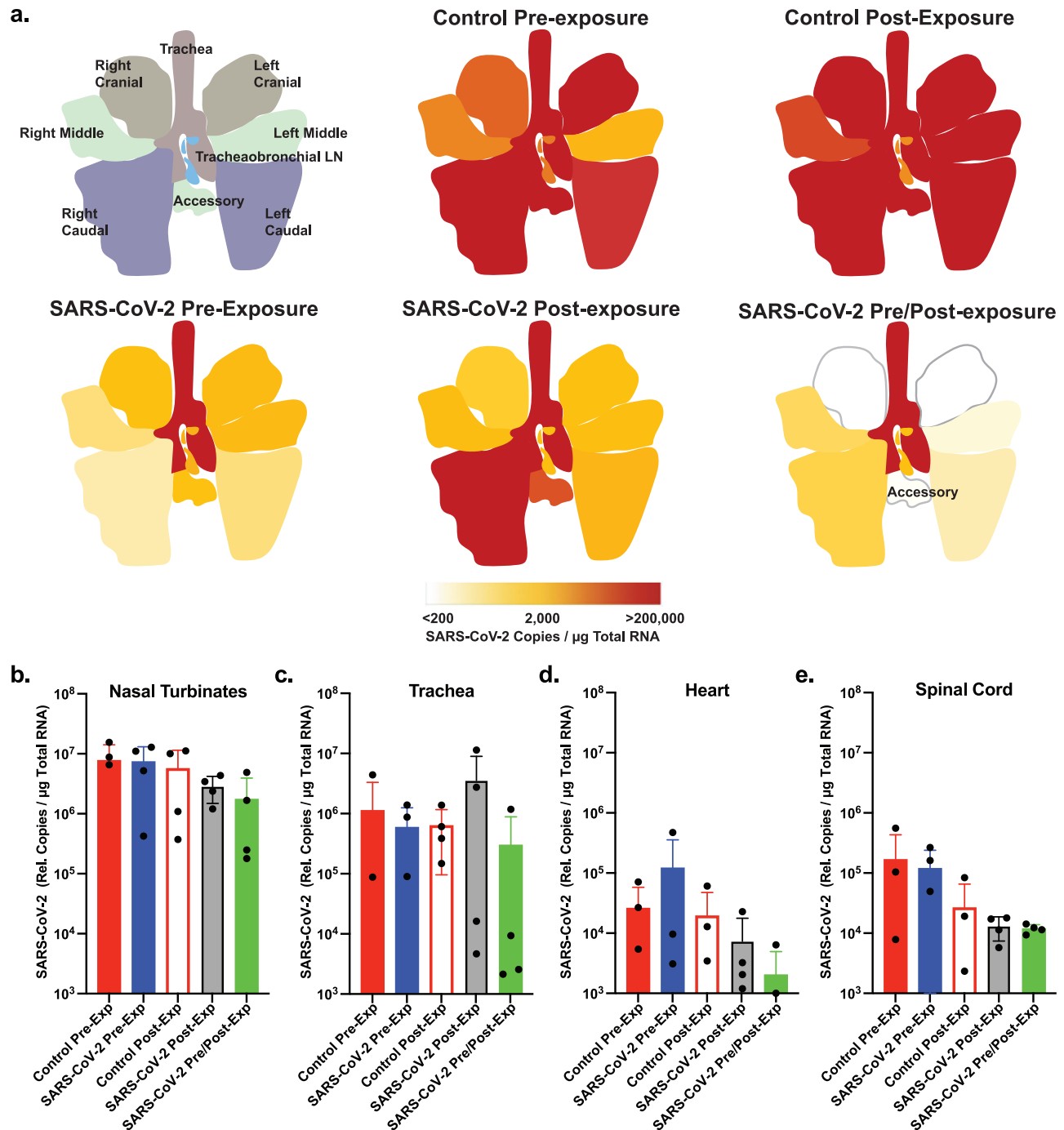

**Fig. 3 | Inhaled aerosolized anti-SARS-CoV-2 mAb treatment blocks SARS-CoV-2 replication in lung tissues and limits viral spread. a** Graphical depiction of lung anatomy (upper left) and SARS-CoV-2 RNA detection levels in respiratory tract tissues. The graded scale shows the average relative copy number for each group (n = 4/group). Caudal lobes consistently have the highest level of virus detection across groups, but the SARS-CoV-2 mAb therapy reduces viral RNA levels relative to the control RSV mAb. **b–e** RNA was extracted from respiratory tract, heart and spinal cord tissues, and viral RNA levels were determined by qRT-PCR using SARS-CoV-2 specific primers and probes. Individual data points for each animal within each group (n = 4 biologically independent animals per group) are shown as are the mean and SEM for each group. Colors in the graphs are: solid red, Control Pre-Exp group; solid blue, SARS-CoV-2 Pre-Exp group; open red, Control Post-Exp group; gray, SARS-CoV-2 Post-Exp group; and solid green, SARS-CoV-2 Pre/Post-Exp group. Source data are provided as a Source Data file.

dysregulate type I interferon (IFN), inflammatory and T cell-mediated responses, and to stimulate the subsequent cytokine storm, leading to lung damage and lymphopenia[5–7]. The cytokine data measured in BAL samples over time showed that 12 inflammatory cytokines were upregulated following infection in animals in the control groups, while the SARS-CoV-2 mAb post-exposure group demonstrated upregulation of only a few of these cytokines. Unexpectedly, we observed that animals treated with post-exposure AR-701, which were the least able

to reduce viral RNA or infectious virus in BAL, displayed an increase in CXCL9, CXCL10, which attract activated effector T lymphocytes and NK cells in a process that may help clear the infection and promote wound healing but also have been implicated in increased pathogensis. Taken together, these data indicate that SARS-CoV-2-infected NHPs treated with aerosolized mAbs targeting the virus both pre-exposure and pre/post-exposure were protected against inflammatory cytokine increases, and all treated animals had reduced lung pathology. The

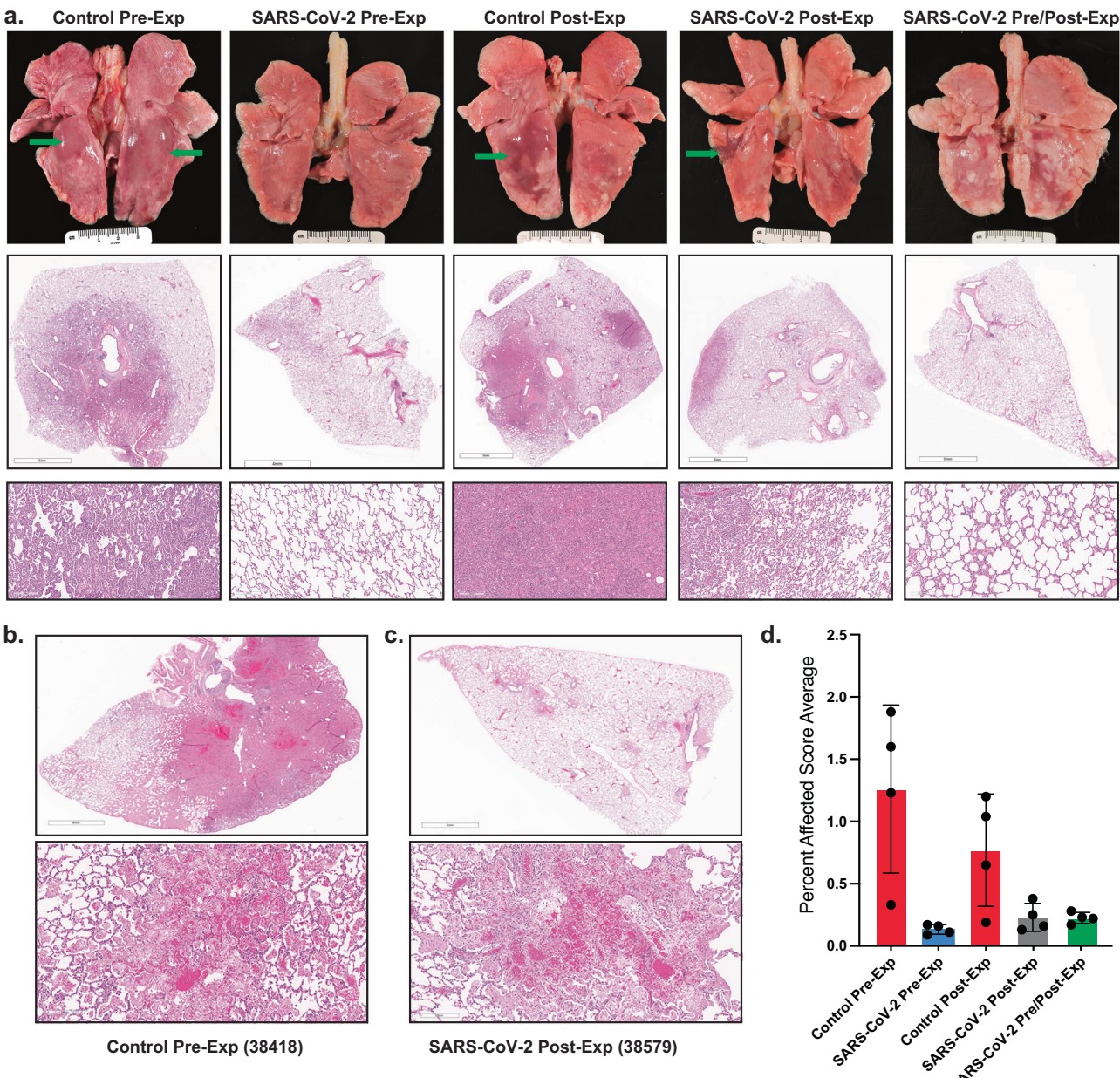

**Fig. 4 | Inhaled aerosolized anti-SARS-CoV-2 mAb treatment prevents extensive lung tissue damage. a** Top row shows photographs of lungs at necropsy for representative animals from each of the 5 treatment groups. The additional rows of micrographs show representative tissue sections stained with hematoxylin and eosin (H&E) at the sub-gross level with higher magnification depicted below. The control mAb pre-exposure group had acute interstitial pneumonia (Grade 4 affecting ~50% of tissue section) with lung lobes mottled light and dark pink with focally extensive pulmonary edema in caudal lobes (green arrows) and regional lymphadenopathy. SARS-CoV-2 mAb Pre-exposure group had minimal interstitial pneumonia (Grade 1 affecting ~10% of tissue section) with mildly mottled light and dark pink lung lobes and mild regional lymphadenopathy. Control mAb post-exposure group exhibited moderate to severe interstitial pneumonia (Grade 4 affecting up to 50% of tissue section) with multiple foci of non-collapsing parenchyma most commonly observed in the caudal lung lobes (green arrow). SARS-CoV-2 mAb post-exposure group had minimal to moderate interstitial pneumonia (Grade 3 affecting ~10% of tissue section) with mottled light and dark pink areas

present most abundantly in the caudal lung lobes (green arrow) with regional lymphadenomegaly. SARS-CoV-2 mAb Pre/Post-exposure group had minimal to mild interstitial pneumonia (Grade 2 affecting ~10% of tissue section) with mottled light and dark pink lung lobes that affected the caudal lung lobes primarily. Scale bar = 5 mm in middle panel photomicrographs except for the SARS-CoV-2 Pre-exposure group, which = 4 mm. Scale bar in bottom panel = 200 μm. **b** and **c** H&E staining of lung sections. Scale bars = 4 mm and 200 μm in the upper and lower images, respectively. **b** Necrotizing, fibrinous hemorrhagic pneumonia detected in over 50% of the tissue section from animal 38418 (control mAb pre-exposure). **c** Animal 38579 (SARS-CoV-2 mAb post-exposure), demonstrated necrotizing, fibrinous hemorrhagic pneumonia affecting ~10% of tissue section.
**d** Semiquantitative scoring system was developed by assessing the interstitial cellularity of the alveolar septa as illustrated in Supplementary Fig 4. Individual data points for each animal (n = 4 animal per group) within each group are shown as the mean and SEM for each group.

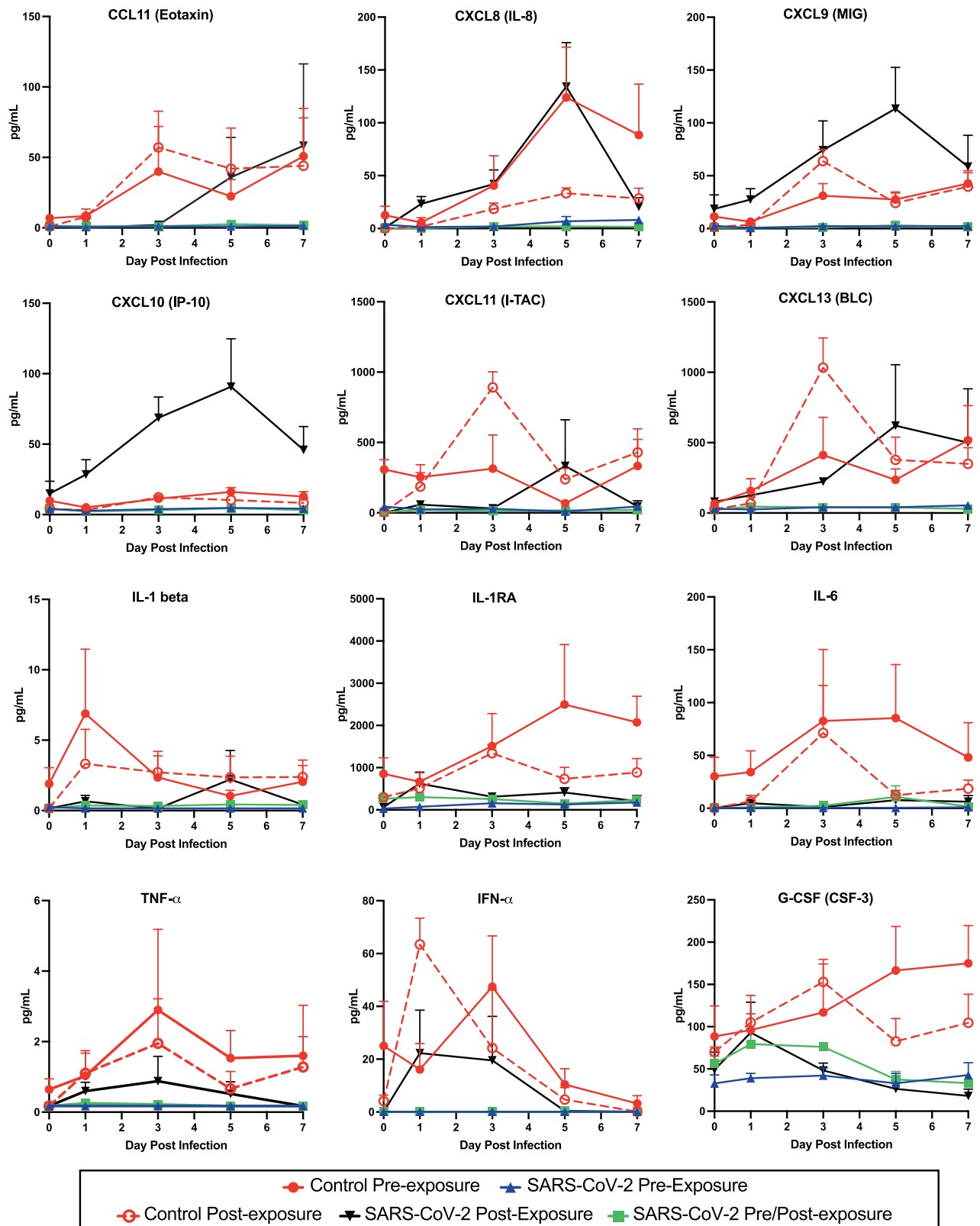

**Fig. 5 | Pulmonary immune activation following SARS-CoV-2 infection is blocked by inhaled aerosolized anti-SARS-CoV-2 mAb treatment.** Kinetic analysis of chemokine and cytokine production was quantified for clarified BAL samples using Luminex bead-based assays. Each graph represents data for one chemokine or cytokine, as labeled at the top of the graph. A protein standard curve was generated from a 7-point dilution series and used to calculate concentration. Individual data points for each animal (*n* = 4 individual animals per group) within each group are shown as the mean and SEM for each group at each timepoint. Colors in the graphs are: solid red circle, Control Pre-Exp group; solid blue triangle, SARS-CoV-2 Pre-Exp group; open red circle, Control Post-Exp group; black triangle, SARS-CoV-2 Post-Exp group; and solid green square, SARS-CoV-2 Pre/Post-Exp group. Source data are provided as a Source Data file.

dual effects of antibody treatment in reducing both inflammatory cytokines and viremia in this rhesus macaque model suggest that antibodies delivered by the aerosol route would be similarly beneficial in humans.

There are several potential advantages to using mAbs to treat respiratory diseases. Treatment with the monoclonal palivizumab has been shown to be safe and to reduce wheezing associated with RSV when given to healthy preterm infants who are at high risk of pulmonary damage due to severe RSV infection[37]. Inhalation delivery was well tolerated and should provide additional benefits to antibody-based therapies for the diseases that target the respiratory tract. First, high concentrations of neutralizing mAbs can be applied directly and immediately to upper and lower respiratory tract mucosal surfaces at high delivery efficiency. A significant dose-sparing effect is expected, potentially resulting in substantial cost savings. The particle size can be controlled by nebulizing devices, so that droplets less than 4 microns in size can most effectively reach the lower respiratory tract. Antibodies, in particular IgG, are stable to nebulization from commercially available nebulizers (jet and vibrating mesh nebulizers). Secondly, inhaled mAbs could be self-administered on an outpatient basis or at home, thus promoting earlier treatment and avoiding the need for face-to-face interactions when virus is actively shed and social distancing is necessary. Although mAbs were rapidly cleared from the upper airway, resulting in less control of viral replication in this compartment, at-home delivery systems may allow for more frequent administration and overcome this drawback. In the future, both availability and delivery could be facilitated by the use of extremely stable, lyophilized or spray dried antibodies that could be solubilized and delivered via nasal sprays or delivered directly from dry powder inhalers, respectively. One of the most significant cost drivers for human pharmaceuticals is the need to ship and store products in the cold chain to assure stability. If antibodies could be freeze or spray dried until use, the requirement for the cold chain would be eliminated. Finally, delivering mAbs directly to the respiratory tract could be an effective way to block viral shedding and person-to-person transmission.

Nonetheless, there are drawbacks to the use of antibodies as therapies, given their exquisite specificity and loss of efficacy as new variants emerge[20,38]. It will be critical to identify antibodies that are potent against nearly all variants, such as the pan-sarbecovirus neutralizing antibody AR-703 mAb used in this study or others[39]. With these antibodies in hand, there are cases where parenteral delivery of mAbs may be indicated, particularly if early aerosolized antibody intervention is not possible and it is suspected that the virus has already spread from the respiratory tract to other organ systems. In many other situations, however, aerosol delivery could be much more widely available and may be a future solution. The remaining issues for widespread aerosol delivery involve availability, delivery methods, and timing. Our study indicates that a combination of pre- and post-exposure delivery of SARS-CoV-2 antibodies was the most effective against a very high viral challenge in primates. For many viral diseases, pre-exposure prophylaxis is not practical. But for SARS-CoV-2 there is experience in recognizing high-risk exposure situations, and individuals could self-administer an aerosol preparation and pre-arm themselves prior to attending crowded events or other high-risk places. Repeated post-exposure self-administration of mAbs to prevent COVID and other respiratory diseases could also be a possible approach for the future.

## Methods
### Study design and animal procedures
This study included 22 (12 male, 10 female) juvenile (2–3 years old) Indian-origin rhesus macaques (*M. mulatta*) ranging in weight from 3.5–5.65 kg at the study endpoint. The animals were housed at the Oregon National Primate Research Center (ONPRC) in an ABSL-2 facility for non-infectious studies and in an ABSL-3 biocontainment facility for all SARS-CoV-2 in vivo work. All macaques in this study were managed according to the ONPRC animal care program, which is fully accredited by AAALAC International and is based on the laws, regulations, and guidelines set forth by the United States Department of Agriculture (e.g., the Animal Welfare Act and Regulations), Institute for Laboratory Animal Research (e.g., Guide for the Care and Use of Laboratory Animals, 8th edition), and the Public Health Service Policy on Humane Care and Use of Laboratory Animals. All animal and laboratory work was reviewed and approved by the Oregon Health and Science University (OHSU) Institutional Biosafety Committee (IBC) and the OHSU West Campus Institutional Animal Care and Use Committee (IACUC). All animals were specific pathogen-free and tested negative for SARS-CoV-2 prior to inclusion in the study. Animals were socially housed or provided protected contact with conspecifics during the study. Sex as a variable was not considered as part of the study design.

The study design for the SARS-CoV-2 infection group is depicted in Fig. 1a, and the breakdown of animal groups is listed in Table 1. Animals were infected via intratracheal and intranasal delivery[23] of virus inoculum containing $1 \times 10^6$ pfu of SARS-CoV-2 Delta (BEI Resources, Isolate hCoV-19/USA/MND-HP05647/2021 B.1.617.2; Delta Variant, WCCM; Catalog No. NR-56116; Lot 70047614) diluted in phosphate-buffered saline. The challenge stock virus was subjected to deep sequencing to identify mutations (Supplementary Fig. 7). For viral infections, the animals were sedated with ketamine and placed in dorsal recumbency. Using a laryngoscope, a sterilized 5 or 8 Fr tube was passed into the trachea to the level of the manubrium. The viral inoculum, contained in a 3 mL syringe, was delivered and then followed by 1–2 mL of air to evacuate the tube. The animals were then immediately inoculated intranasally. The plastic sheath of a 24 g catheter was connected to a syringe containing a 1 mL inoculum. The flexible catheter sheath (~1 inch long) was gently guided into the nares up to the hub prior to dispensing the inoculum, which was divided equally between the nostrils. The head and chest of the animal were then elevated for 10-15 seconds. All blood collection and mucosal swab procedures were performed under ketamine sedation. Bronchoalveolar lavage (BAL) was performed under sedation with a combination of ketamine and dexmedetomidine. Peripheral blood mononuclear cells and plasma were isolated using lymphocyte separation medium. BAL fluids were collected using a bronchoscope and the samples were clarified by centrifugation. The BAL fluid was analyzed for mAb concentration, viral loads and cytokine secretion. Nasal swabs were collected using a Copan FLOQSwab nasal swab (Thermo Fisher Scientific #23-600-952) that was inserted into the nares, gently rotated, and then immersed in 0.4 ml of PBS (containing 1x protease inhibitor cocktail; Thermo Fisher Scientific #78415) or Dulbecco's Modified Eagle's Medium (DMEM) containing 1× penicillin–streptomycin-Glutamine (Thermo Fisher Scientific #10378016) and 1× Antibiotics-Antimycotic solution (Thermo Fisher Scientific #15240062). Oropharyngeal swabs were collected using BVI Weck-Cel Spears (Thermo Fisher Scientific #NC0240640) by placing swabs on the oral mucosa in the caudal oropharynx for approximate 5-10 seconds then placed in the same media as described previously for nasal swabs. Swabs were removed and samples were processed for vRNA extraction and detection, viral focus-forming assay, or mAb detection. All animals were euthanized 7 days after inoculation with SARS-CoV-2 and complete necropsies were performed.

### Inhaled delivery and lung-deposited dose estimates
The optimized Investigational Nebulizer System (https://www.pari.com/int/eflow-technology-partnering/) was used for delivery of aerosolized antibodies to macaques sedated with ketamine. The device includes a port to enable simultaneous oxygen and aerosol delivery to the animals. A similar optimized device derived from the eFlow-Technology platform has been shown previously to deliver 2.0 mL

(100 mg) of a 5% IgG solution (50 mg/mL) to macaques in 2.2 min[29]. The optimized eFlow device used for this study produces particles with a mass medium diameter of 3.6 μm that have been shown to reach the lower lung of macaques and humans. Delivery was accomplished using a silicon facemask (PARI Smartmask Baby/kids) for human infants that was fitted for an approximately 4 kg rhesus macaque and positive oxygen flow of 2 liters/minute. The mask permitted the deposition of antibody in the nasal cavity, oropharynx, and lung and the exhaled $CO_2$ to escape from the nebulizer through the nose and mouth. We determined an approximate delivered dose of antibodies by measuring mAbs in BAL fluid and mucosal swabs longitudinally. The animals were allowed to breathe in the entire dose of aerosolized antibodies, which took up to 10 min. Animals were monitored for heart rate, respiratory rate, blood oxygen saturation via pulse oximetry, body temperature, and mucus membrane color during nebulization while sedated.

The dose from the delivery device that the animals inhaled (inhaled dose) and the lung-deposited dose are a fraction of the dose that is loaded into the nebulizer device and are generally estimated based on device geometry and performance parameters. The volume of liquid that is retained in the device reservoir plus the liquid that can accumulate in the device chambers and tubing (referred to as rainout or holdup liquid) are estimated be 0.5 mL or less. Of the remaining aerosol that reaches the mask ~50% will be lost from the exhalation port of the mask during the exhalation phase of the macaques breathing pattern, resulting in an estimated delivered (or inhaled) dose of approximately 40% of the device-loaded dose. For an aerosol droplet size of 3-4 μm generated by the eFlow-Technology nebulizer, a lung deposition fraction of 15% of the inhaled dose is estimated for nonhuman primates[40]. Thus, approximately 6% of the device dose was estimated to reach the lungs for macaques used in these studies.

## Isolation and specificity of monoclonal antibodies

Four mAbs cloned from human subjects and expressed as IgG1s were purified and characterized for in vitro binding and antiviral activity for this study: three directed to the SARS-CoV-2 S-protein and a fourth antibody directed to the RSV fusion protein called 25P13[34]. CoVIC-96 was isolated from a convalescent COVID-19 patient who had high titers of neutralizing antibodies; it potently neutralizes the Delta variant of concern ($IC_{50}$~3 ng/mL) and targets the epitope community termed RBD-5[33]. Antibodies in the RBD-5 epitope community bind on the outer face of the RBD. Like most RBD-5 antibodies, CoVIC-96 lost neutralization activity against Omicron and its sublineages[41]. AR-701 is a combination of two human immunoglobulin G1 (IgG1) mAbs discovered from screening the B-cells of convalescent SARS-CoV-2 infected (COVID-19) patients. Each mAb of the AR-701 cocktail neutralizes coronaviruses using a distinct mechanism of action, namely inhibition of viral fusion and entry into human cells (AR-703)[27] or blockage of viral binding to the human ACE2 receptor (AR-720)[28]. The activity of the two mAbs complement and enhance each other in a synergistic fashion, creating a potent combination. AR-720 (also called elsewhere 1213H7 mAb) binds to the RBD of the S protein of SARS-CoV-2 in epitope community RBD-2b[33,42], while AR-703 (also called elsewhere 1249A8) binds to the S2 stalk stem helix region of S proteins from betacoronaviruses, including many SARS-CoV-2 variants (Beta, Gamma, Delta, Epsilon, and Omicron)[27]. Both mAbs bind to the Omicron subvariants, BA.1, BA.2, BA.4, and BA.5 with comparable affinity compared to the original Wuhan strain. All authentic live SARS-CoV-2 Beta, Gamma, Delta, Epsilon, and Omicron variants, SARS, and MERS-CoV tested were neutralized in vitro by the AR-701 mAb combination. Multiple animal challenge models widely used to evaluate COVID-19 treatments support the broad efficacy of AR-701 against the original Wuhan wild-type strain, the Delta variant, the Omicron variant, and SARS-CoV. The AR-701 mAbs are engineered in the Fc regions to prolong their half-life and be active for 6–12 months in the blood by the insertion of the YTE mutations[43]. A similarly prepared human IgG1 mAb directed to the RSV fusion protein was chosen for use as a control mAb. The dose ratio of AR-703 to AR-720 in the AR-701 cocktail was 2:1.

## Antibody production and purification

CoVIC-96 and the RSV mAb 25P13 were produced under contract at Zalgen. AR-703 and AR-720 mAbs were produced under contract to Aridis Pharmaceuticals. Antibody expression vectors were transfected into *ExpiCHO* cells. Abs were purified from the supernatant by affinity chromatography and formulated in 8% sucrose, 10 mM citrate, 0.4% polysorbate-80, pH 6.5. Purity was assessed by SDS-PAGE and shown to be greater than 95% heavy and light chain. Endotoxin levels were less than 0.125 EU/mL. Specificity of CoVIC-96 was determined by binding to full length SARS-CoV-2 Spike protein expressed in HEK-293T/17 cells and lack of binding to cells transfected with vector control. Prior to any in vivo work, activity and specificity of each mAb (AR-703, AR-720 and 25P13) were confirmed by antigen-specific ELISA and SARS-CoV-2 focus-forming assays described below. Cell lines were not authenticated for this experiment.

## Quantification of mAbs

To measure nebulized SARS-CoV-2 and RSV mAbs in plasma, BAL, and secretions, enzyme-linked immunosorbent assays (ELISA) were performed largely as described by Malherbe et al.[44]. Briefly, half-well ELISA plates (Costar) were coated with recombinant RSV protein or COVID-19 Delta spike protein (HexaPro; expressed and purified as described in Hastie et al.[33]) by incubating 40 ng/well in 0.2 M $H_2CO_3$ buffer pH 9.4 at 4 °C overnight. Plates were then washed in binding buffer (PBS pH 7.4 + 0.1% Triton X-100) and blocked with 150 μL PBS containing 5% dried milk and 1% goat serum for 1 h at room temperature. Blocking buffer was discarded and threefold serial dilutions of plasma or control antibodies were added to unwashed wells in 50 μL binding buffer. The anti-RSV mAb was used as a negative control. The anti-SARS-CoV-2 IgG mAb (Aridis) was used to create a standard curve. After 1 h at room temperature, plates were washed 3x and then probed for 1 h with 50 μL of a 1:5000 dilution of a horseradish peroxidase-conjugated goat anti-human IgG F(ab)2 fragment-specific polyclonal antibody (Jackson ImmunoResearch catalog no. 109-035-006). Plates were then washed 5×, and bound Ab was visualized by the addition of 50 μL tetramethylbenzidine (BioFX) for 10 min before stopping the reaction with 50 μL 1 N $H_2SO_4$. Optical density (450 nm) was measured immediately on a SoftMax® Pro 5 microplate reader (Molecular Devices). mAb concentrations in all samples were quantified after determining the 50% effective binding ($EC_{50}$) by nonlinear regression in GraphPad Prism v 9.0 for Mac and Office Excel. All ELISA assays were performed in duplicate and repeated at least two times.

## Protein analysis by SDS-PAGE

SDS-PAGE was performed in 4–12% gradient gels according to manufacturer's specifications (Thermo Fisher Scientific). Briefly, samples were mixed with 4× NuPAGE sample buffer and 10× NuPAGE reducing agent then heated at 70 °C for 10 min. Samples were cooled and centrifuged at 14,000 × $g$ for 5 min before transfer into gel wells. PageRuler Prestained Protein Ladder Plus (Fermentas) was included as a molecular weight standard. The gel was run at a constant 200 V for 50 min in 1× NuPAGE MOPS running buffer. The gel was rinsed in $dH_2O$, and the resolved bands were disclosed with Novex SimplyBlue colloidal stain.

## Sequencing of SARS-CoV-2 RNA

To identify sequence mutations within the viral stock and some representative tissues from necropsy, isolated viral RNA was subjected to first strand synthesis reverse transcription (RT) to produce single-stranded cDNA using LunaScript RT SuperMix Kit (NEB). Viral sequence was then amplified via pooled amplicon PCR using a 1200 bp

overlapping amplicon strategy developed by Freed et al. (a.k.a. midnight primer set, IDT)[45]. Individual specimen PCR reactions were pooled, cleaned, and then subjected to shotgun sequencing library preparation utilizing the hyperactive Tn5 transposase to simultaneously fragment target DNA and appends sequencing adapters in a single step in a processes referred to as '

tagmentation (commercially available as the Nextera product)[46]. To enable sample multiplexing, we have modified a tagmentation strategy that has been developed for single-cell genomics assays (s3)[47]. A set of uniquely indexed Tn5 molecules generated in-house is used, allowing for PCR products from 96 different specimens to be pooled after Tn5 addition for a single indexing using standard Illumina sequencing chemistry. Index PCR products were cleaned using SPRI beads and analyzed on a D1000 high sensitivity tapestation (Agilent). Samples were sequenced on a NextSeq2000. FASTQ reads were quality trimmed and aligned to the sequence of the MD-HP05647 challenge stock, using BWA-mem. Duplicates molecules were marked and sequences terminating in the primer regions were removed. Quasispecies variant calling was performed using LoFreq, a variant caller designed for accurate calling of rare variants, which is well suited to viral quasispecies[48]. These analyses provide a table of nucleotide variants per sample, with the associated read depth and frequency within the viral quasispeces. Only two mutations were present in more than 5% of the challenge stock quasispecies: 15953:C>T in 9.8% of reads and 26668:C>T in 6.4% of reads (Supplmentary Fig. 7a).

## Quantification of SARS-CoV-2 RNA

RNA was extracted from SARS-CoV-2-infected rhesus macaque tissue, BAL, NAS and OP samples. Briefly, 300 μL of resuspended NAS and OP samples and 300 μL BAL fluid were processed with the Maxwell 48 sample RSC automated purification system using the Maxwell RSC Viral TNA extraction kit (Promega # AS1330). Total nucleic acids were resuspended in 60 μl RNase-free water. Rhesus macaque tissue samples were homogenized in 1 mL Trizol reagent (Thermo Fisher # 15596026) and centrifuged for 5 min at 5000 × $g$. The clarified sample (~200 μL) was then processed with a Directzol RNA Miniprep Plus kit (Zymo Research # R2070) following the manufacturer's instructions and resuspended in 30 μL RNAse-free water. The isolated tissue RNA samples was quantified using a Nanodrop spectrophotometer and diluted to 100 ng/μl in RNAse-free water. To detect viral genomes, real-time PCR was performed using primers and probes specific for the N region of SARS-CoV-2, CDC N2 primer and probe set: 2019-nCoV_N2 Forward TTACAAACA TTGGCCGCAAA, 2019-nCoV_N2 Reverse GCGCGACATTCCGAAGAA, and FAM-ACAATTGC CCC CAGCGCTTC AG- BHQ-1 (Integrated DNA Technologies). For BAL, nasopharyngeal and oropharyngeal samples, 5 μL of total RNA was added to 10 μL Taqpath 1-Step RT Master mix CG (Thermo Fisher # 15299) in triplicate and run on a QuantStudio 7 Real-time PCR machine. Tissue total RNA (~5 μL) was assayed as above in triplicate, and all data was analyzed using QuantStudio 6 and 7 Flex real-time PCR System software. Detection of viral sub-genomic RNA was determined using Real-Time PCR with primer and probes specific for sub-genomic viral RNA, sgLeadSARSCoV2_F CGATCTCTTGTA-GATCTGTTCTC, wtN_R4 GGTGAACCAAGACGCAGTAT and probe 6-FAM-TAACCAGAA-ZEN-TGGAGAACGCAGTGGG-IBFQ (Integrated DNA Technologies) as previously described[49]. Real-time PCR detection of sgmRNA was performed as described above. The limit of detection for this assay is ~50 copies.

## SARS-CoV-2 focus-forming assays

Vero cells expressing transmembrane protease, serine 2 and human angiotensin-converting enzyme 2 (Vero E6-TMPRSS2-T2A-ACE2; obtained through BEI Resources, NIAID, NIH, NR-54970) were plated onto 96-well tissue culture dishes. Cell lines were not authenticated at our institution. BAL fluid or NAS/OP swabs in PBS were serially diluted five-fold in Dulbecco's Modified Eagle's Medium (DMEM)

supplemented with 5% fetal bovine serum and antibiotic-antimycotic (streptomycin, penicillin, amphotericin B). Diluted samples were added to cells in a total volume of 50 μL, incubated for 1 h at 37 °C, 5% $CO_2$, and then overlaid with 0.5% carboxymethyl cellulose in culture medium. At 24 h after infection, cells were fixed with 4% paraformaldehyde, washed twice with PBS and blocked/permeabilized for 1 h in PBS supplemented with 2% normal goat serum (NGS; Sigma) and 0.4% Triton X-100. Cells were then washed twice with PBS followed by incubation with 0.25 μg/mL anti-SARS-CoV-2 monoclonal antibodies recognizing the N (Genetex, GTX635712) or S (Genetex, GTX632604) proteins in PBS supplemented with 2% NGS for 1 h, washed twice more with PBS, incubated with anti-mouse IgG-horseradish peroxidase (Rockland) for 1 h, and washed twice with PBS. Foci were visualized by incubation with the Vector VIP peroxidase substrate kit (Vector Labs) according to manufacturer's specifications and counted using an ELIspot reader (AID).

## Analysis of tissues at necropsy

Representative tissue samples from the major organs, tissues in the respiratory tract, spinal cord, lymph nodes, spleen, and brain were collected and stored in RNAlater, Trizol (RNA isolation), and 10% buffered formalin (histological evaluation). After lung tissue procurement and photography, each lung lobe edge was clamped to allow removal of a tissue slice for RNA processing for virologic determinations. While still clamped, the lung lobes were slowly infused with neutral buffered 10% formalin. Once fully inflated, the main bronchus was tied off, and the lungs were placed in individual jars of formalin and fixed for 72 h. The lungs were then sliced from the hilus towards the periphery into approximately 5 mm-thick slabs.

Tissues collected for microscopic examination were fixed in 10% buffered formalin before embedding in paraffin and production of 5 μm-thick sections for hematoxylin and eosin staining. For lung pathological assessments, nine lung slides, which included two slides of the caudal lobes and one each of the other lung lobes, were scanned at 40× with an Aperio AT2 Leica Biosystems microscope slide scanner and examined by two board-certified veterinary pathologists (ADL; LMAC) blinded to animal group assignments. A semiquantitative scoring system was developed by assessing the interstitial cellularity of the alveolar septa. An initial score reflecting the most severely affected area was assigned to each lung lobe evaluated. The scoring system was: 0 = Normal cellularity; 1 = 1–2 cells thick; 2 = 2–4 cells thick; 3 = 4–6 cells thick; and 4 ≥ 6 cells thick or necrohemorrhagic lesions. Scores are illustrated in Supplementry Fig. 4[23]. The percentage of the tissue exhibiting any degree of increased cellularity and inflammation was estimated. Final scores for each lung lobe were obtained by multiplying the initial scores by the percentage of the tissue sections affected. If the percentage of the lung affected was less than 10%, a final score of 0 was assigned. An average score for each animal was calculated by combining the final scores of all lung lobes divided by the number of slides evaluated. The calculation of the final scores for each animal was based on the methods described and criteria illustrated in Supplementary Fig. 4.

## Cytokine assays

Macaque cytokine assays were performed using a Cytokine/Chemokine/Growth Factor 37-plex NHP ProcartaPlex Panel for the Luminex platform (Thermo Fisher) according to the manufacturer's instructions using a 7-point standard curve. Briefly, BAL samples were clarified by centrifugation and incubated with magnetic beads for 2 h. The beads were washed twice, incubated with detector antibody mix for 30 min and washed twice before incubation with PE-conjugated streptavidin for 30 min. After the incubation, the beads were washed twice and resuspended in reading buffer. Signals were quantified using a Luminex 200 Detection system (ONPRC Endocrine Technologies Core).

## Statistical analyses

All statistical analyses and graphing were performed in Prism v7 software (GraphPad Software, Inc.). For viral burden detection experiments, data were log-transformed, and the Mann–Whitney and Dunnett's tests were used to determine significance and area under the curve statistical analysis was used to assess total burden. For histological results, cytokine analyses, and Šidák's multicomparison test were used to determine significance and area under the curve statistical analysis with a Tukey one-way ANOVA was used to assess total cytokine levels.

## Reporting summary

Further information on research design is available in the Nature Portfolio Reporting Summary linked to this article.

## Data availability

Source data are provided with this paper.

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

## Acknowledgements

We acknowledge the animal care staff at the Oregon National Primate Research Center who provided care for the animals and research support for the work presented in this manuscript. We thank R. Jensen-Reinhardt for project and meeting coordination. We thank Allie Meristem for pathology data collation and whole slide scanning. SARS-Related Coronavirus 2, Isolate hCoV-19/USA/MD-HP05647/ 2021 (Lineage B.1.617.2; Delta Variant) (WCCM), NR-56116, contributed by Andrew S. Pekosz, was obtained through BEI Resources, NIAID, NIH. MAbs were produced and purified at Zalgen or at Aridis. Funding was provided by NIH Oregon National Primate Research Center Core grant P51 OD011092 (P. Barr-Gillespie, N.L.H.), NIH COVID research supplement P51 OD011092-S3 (N.L.H.), R01 AI161175 (J.J.K, L.M.-S, M.R.W.), VA Merit Award I01BX005469 (M.M.S.), COVID-19 Therapeutics Accelerator INV-006133 (E.O.S.), the Bill and Melinda Gates Foundation OPP1210938 (E.O.S.), a supplement to NIH grant U19 AI142790-S1 (E.O.S.). and the GHR Foundation (E.O.S.). Funding for SARS-CoV-2 viral sequencing was provided by the OHSU Foundation (25320, B.O.).

## Author contributions

Conceptualization and experimental design: D.N.S., A.J. Hirsch., J.J.S., H.S., S.L.S, E.O.S., J.E.C., K.K.A.V., P.L., V.T., D.N.F, N.L.H.; Experimental work: D.N.S., A.J. Hirsch, J.J.S., A.D.L., L.C., C.N.K, J.L.S., W.F.S., D.C., J.W., C.N.L, S.N.A., T.R.H.,R.H.C., L.M.B.; Data interpretation and analyses: D.N.S., A.J. Hirsch, J.J.S., A.D.L., L.C., A.J. Hessell, B.J.B., C.N.L., S.N.A., B.J.O., R.J.R., K.J.O., E.O.S., S.L.S., R.H.C., L.M.B., J.E.C., P.L., V.T., D.N.F., N.L.H.; Materials and devices: M.M.S., Z.R.T., M.R.W., L. M.-S., J.J.K., E.O.S., S.L.S., L.M.B., J.K., J.E.C., V.T.; All authors were consulted in the writing of the manuscript and all have approved this version; Major writing: D.N.S., A.J.Hirsch, J.J.S., A.D.L., L.C., A.J. Hessell, B.N.B, E.O.S., S.L.S., J.E.C., K.K.A.V, V.T, D.N.F, N.L.H.

## Competing interests

M.R.W., L.M.-S., and J.J.K. are co-inventors on patents that include claims related to the mAbs described. L.M.B. is Co-Founder and Managing Director of Zalgen Labs. He receives remuneration from the company. The remaining authors declare no competing interests.

## Additional information

¹Vaccine & Gene Therapy Institute, Oregon Health & Science University, Beaverton, OR, USA. ²Oregon National Primate Research Center, Oregon Health & Science University, Beaverton, OR, USA. ³Aridis Pharmaceuticals, Los Gatos, CA, USA. ⁴Department of Molecular and Medical Genetics, Oregon Health & Science University, Portland, OR, USA. ⁵Environmental Health & Safety, Oregon Health & Science University, Portland, OR, USA. ⁶Baltimore VA Medical Center, VA Maryland Health Care System, Baltimore, MD, USA. ⁷Division of Clinical Care and Research, Institute of Human Virology, University of Maryland, Baltimore, MD, USA. ⁸Department of Microbiology, University of Alabama at Birmingham, Birmingham, AL, USA. ⁹Texas Biomedical Research Institute, San Antonio, TX, USA. ¹⁰Department of Medicine, Division of Infectious Diseases, University of Alabama at Birmingham, Birmingham, AL, USA. ¹¹California National Primate Research Center, University of California, Davis, CA, USA. ¹²Center for Infectious Disease and Vaccine Research, La Jolla Institute for Immunology, La Jolla, CA 92037, USA. ¹³Vanderbilt University Medical Center, Nashville, TN, USA. ¹⁴PARI Pharma GmbH, Starnberg, Germany. ¹⁵Zalgen Labs, LLC, Frederick, MD, USA. ¹⁶University of California, Irvine, School of Medicine, Irvine, CA, USA. ✉e-mail: truongv@aridispharma.com; dnfortha@hs.uci.edu; haigwoon@ohsu.edu

