## [Peer Review File · Nature Communications]

Aerosol Delivery of SARS-CoV-2 Human Monoclonal Antibodies in Macaques Limits Viral Replication and Lung PathologyReviewers' Comments:

Reviewer #1:

Remarks to the Author:

Streblow et al present an indepth study of administration of a monoclonal antibody cocktail by aerosol route to NHPs infected with SARS-CoV2. The premise of treatment is very significant as a relatively low cost nasal spray with neutralizing mAB may be an effective mean of controlling Sarbecovirus outbreaks without the need for involvement of healthcae facilities.

The authors have analyzed the level of antibody delivered to the lung and nasopharungeal mucosa. Low levels of antibodies can be detected, which is not surprising given the constraints of delivering nebulated material through face mask to anesthetized animals. It can only be expected that the delivery in humans will be much more efficient. Despite low level of antibody detected the pathology outcomes are impressive. This is despite the fact that the effect on viral RNA in easily accessible samples like BAL, NAS, and OP; i.e. samples that are analyzed in most studies, was less impressive. But the antibody treatment had profound effects in the pathology of the lung, infiltration of immune cells, outcomes tha can be measured only in such an in depth and invasive study. The findings are novel given the limited number of reports on the delivery of nebulized antibodies for treatment of respiratory infections.

There are also missed opportunities here. Given the extensive tissue samples collected it is not clear why the authors did not measure mAB concentration in the lung tissues. It is clear from the plasma concentration that very little mAB is crossing the lung tissue into blood, likely because the antibodies are mostly concentrated in the lung tissue, which could be the reason for the observed effects.

Some minor to moderate critiques:

1- The authors indicate that because of high variability in titers doing statistical analysis was challenging. Isn't variability exactly why statistics is needed? The authors need to do stats on data in Fig. 2, and supplemental figures 2 and 3 and identify the points where statistical significance is observed.

2- There is a marked increase in levels of chemokines in BAL. The authors need to discuss this more in depth as, beside the point they raised about recruiting NK and T cells that can help clear the virus, it can be also a liability. Increased CXCL8, CXCL9, CXCL10 can be also pathological as shown in multiple diseases.

3- The authors need to clarify if their PK assay can detect antibodies bound to the virus, this would be important for interpretation of the results.

Reviewer #2:

Remarks to the Author:

The authors conducted a controlled experiment with five groups (n=4) to assess the pre-exposure and post-exposure efficacy of an antibody cocktail delivered by inhalation of aerosols generated via nebulization. The primary measures of success (readouts) were 1) viral load measurement (including gRNA, sgRNA and infectious SARS-CoV-2), 2) lung pathology examined grossly, histopathologically and by semi-quantitative scoring, and assessment of proinflammatory cytokines in bronchoalveolar lavage samples collected post-exposure.

The authors show a reduction of the SARS-CoV-2 load (both RNA and infectious virus) in samples, and tissues as well as a reduction of the lung tissue damage assessed on day 7 post-exposure. Overall, the work is original and topical. Although the data are very promising, the conclusions lack statistical support.

Further, it is unfortunate that the data are confounded by the administration of concomitant medication to the nonhuman primates exhibiting clinical signs of disease. There are also shortcomings in the experimental design which could have been avoided.

Major comments

- It is my opinion that analgesic treatment with meloxicam (COX-2 inhibitor) should not have occurred. Concomitant medication must be avoided when evaluating the efficacy of a therapeutic and it is a serious experimental flaw that biases the data collected within the respective groups. If NHPs were sick enough to meet the euthanasia criterion, then they needed to be euthanized. Alleviating discomfort from clinical infection is not acceptable if this potentially compromises the integrity of the scientific data collected. I encourage the authors to discuss this feedback with institutional stakeholders on how to avoid this in their future studies (for example by adjusting the IACUC protocol and USDA pain and distress categories). As the authors state in lines 343-345, the goal was to observe clinical effects using nebulized mAb delivery following virus exposure; unfortunately, this goal must be considered confounded after treating the sickest NHPs with meloxicam on days 5 and 6, even if the data are still considered "in line" with what was generally expected/favored.

- Statistical analysis with n=4 NHPs does not provide sufficient power, and should be replaced with a description of findings instead. It is unfortunate that the authors cannot go back and re-design the study. My wish and suggestion would have been to focus on either the preventative or therapeutic aspect of aerosol mAb delivery (the one of higher importance), and combine the two control groups into one control arm only. Did the authors have a sufficiently high number of historical controls and resulting data on SARS-CoV-2 natural history (untreated)? The anti-RSV mAb control group is important for the author's pathology readout, but the NHPs in this group should have received the mAb not once, but with the highest frequency corresponding with the respective anti-SARS-CoV-2 cocktail. The total number of NHPs (n=20) could have been distributed across two groups total with n=10; one control group exposed to SARS-CoV-2 and receiving three doses of the anti-RSV control mAb (using the same frequency as the test group). The test group (for therapeutic evaluation) could have received the mAb cocktail once pre- and twice post-exposure. Alternatively, a n=7 (preventative), n=7 (therapeutic) and n=6 (control) grouping design would also have been possible and would have provided better power. The goal should have been to show high-powered efficacy data for the administration strategy that is of most relevance.

- Males and female animals needed to be balanced across groups. Why did the authors choose to assign males to Groups 1 and 2 and females to Groups 3 and 4 given that there were equal numbers of animals from both sexes?

Minor comments

- Consider rephrasing the paragraph on vaccines within the introduction, and/or attenuating the limitations of active immunization/vaccines. There are no real alternatives for vaccination as a preventative strategy to boost herd immunity, and it is the most accessible and cost effective way to achieve immunity within a population. The authors propose aerosol delivery of their mAb cocktail predominantly for therapeutic purposes and to lesser extent for a preventative one (or on a smaller scale to protect high-risk individuals), so this approach is in no real "competition" to vaccines. I suggest re-framing this context.

- The result section on the histopathology data is lengthy (descriptive/qualitative histopathology findings) and could be shortened and be more focused.

- Consider rephrasing general statements such as "there is encouraging news", "modulating disease", "right tissues" and "wreaks havoc with lung tissue" for example.

- Consider rephrasing COVID Post-exposure group to Virus post-exposure group. COVID-19 is reserved to describe the disease in humans.

RESPONSE TO REVIEWER COMMENTS

Responses to each point are provided in italics.

Reviewer #1 (Remarks to the Author):

Streblow et al present an in-depth study of administration of a monoclonal antibody cocktail by aerosol route to NHPs infected with SARS-CoV2. The premise of treatment is very significant as a relatively low-cost nasal spray with neutralizing mAB may be an effective mean of controlling Sarbecovirus outbreaks without the need for involvement of healthcare facilities.

The authors have analyzed the level of antibody delivered to the lung and nasopharyngeal mucosa. Low levels of antibodies can be detected, which is not surprising given the constraints of delivering nebulated material through face mask to anesthetized animals. It can only be expected that the delivery in humans will be much more efficient. Despite low level of antibody detected the pathology outcomes are impressive. This is despite the fact that the effect on viral RNA in easily accessible samples like BAL, NAS, and OP; i.e. samples that are analyzed in most studies, was less impressive. But the antibody treatment had profound effects in the pathology of the lung, infiltration of immune cells, outcomes that can be measured only in such an in depth and invasive study. The findings are novel given the limited number of reports on the delivery of nebulized antibodies for treatment of respiratory infections.

The authors appreciate the comments of the reviewer concerning novelty of the approach and the effects upon lung pathology. In the revised manuscript, we have included statistical analyses on the viral RNA and infectious virus titers as well as the cytokine analyses that show significant differences in comparison to the combined controls.

There are also missed opportunities here. Given the extensive tissue samples collected it is not clear why the authors did not measure mAB concentration in the lung tissues. It is clear from the plasma concentration that very little mAB is crossing the lung tissue into blood, likely because the antibodies are mostly concentrated in the lung tissue, which could be the reason for the observed effects.

The authors appreciate this point that antibody concentrations in the lung are of interest. In this study, the goal was to determine the effects of the nebulized antibody on the outcome at the end of 7 days. We chose not to sample lungs during the experiment in order to preserve the integrity of the tissue and to avoid confounding variables due to sampling. We did sample lungs and turbinates/upper airway at the end of the study to measure virus but did not measure mAbs in tissues. We agree that this was a missed opportunity, as our group has measured antibody distribution in tissues after subcutaneous delivery (Hessell et al. 2016). Nonetheless, we do not expect that antibody levels in tissues would have been significant by 7 days, due our data that showed rapid clearance and lack of significant transfer to the serum.

Some minor to moderate critiques:

1- The authors indicate that because of high variability in titers doing statistical analysis was challenging. Isn't variability exactly why statistics is needed? The authors need to do stats on data in Fig. 2, and supplemental figures 2 and 3 and identify the points where statistical significance is observed.

As noted, we agree with the reviewer, and we now show analyses for the three major quantitative assays. RNA samples over time show significant reductions in virus levels with the treatment by 2-way ANOVA. Because this experiment was a test of therapy vs. controls, we combined the controls into a single group and found significance, primarily in BAL samples.

2- There is a marked increase in levels of chemokines in BAL. The authors need to discuss this more in depth as, beside the point they raised about recruiting NK and T cells that can help clear the virus, it can be also a liability. Increased CXCL8, CXCL9, CXCL10 can be also pathological as shown in multiple diseases.

All cytokine data are now included with this submission in the supplementary data tables. The reviewer makes an excellent point about CXCL8, CXCL9, and CXCL10. In this study we did not observe an increase in these cytokines except in the SARS-CoV-2 Post-Exposure group, and these animals were the least able to reduce viral RNA and infectious virus in BAL (not significantly different from the controls). We note that certain chemokines may be recruiting activated T cells, and we have also included a point about potential pathogenesis. We have also added reference to a newly published study showing that while there is impressive reduction of inflammation with baricitinib in primates infected with SARS-CoV-2, this treatment insufficient to reduce viremia¹.

3- The authors need to clarify if their PK assay can detect antibodies bound to the virus, this would be important for interpretation of the results.

All pharmacokinetic (PK) assays were performed in the absence of infection. We understand the importance of determining PK, and that antibodies bound to virions may not be measurable in liquid samples. There are two important aspects of this experiment that provide additional confidence for the PK data. The first is that the primary method for determining the PK of the SARS-CoV-2 antibody cocktails, or individual antibodies, was to perform the quantification in the absence of virus infection. Second, we determined the decay/clearance of the control RSV mAb in vivo in the presence of SARS-CoV-2 infection, after having demonstrated that it does not bind to SARS-CoV-2 antigens. As stated in the Results, in infected animals, when we compare the levels of antibodies detected in the control animals and the treated animals receiving a single dose of mAbs, we observed similar clearance levels, suggesting that virion binding has not appreciably affected the measurement of the SARS-CoV-2 mAbs. During the infection phase, rigorous PK studies could not be performed in the infected animals without compromising the delivery of the antibodies or the health of the animals, because daily sampling was not possible,

and multiple dosing in some animals compromises PK measurements.

Reviewer #2 (Remarks to the Author):

The authors conducted a controlled experiment with five groups (n=4) to assess the pre-exposure and post-exposure efficacy of an antibody cocktail delivered by inhalation of aerosols generated via nebulization. The primary measures of success (readouts) were 1) viral load measurement (including gRNA, sgRNA and infectious SARS-CoV-2), 2) lung pathology examined grossly, histopathologically and by semi-quantitative scoring, and assessment of proinflammatory cytokines in bronchoalveolar lavage samples collected post-exposure.

The authors show a reduction of the SARS-CoV-2 load (both RNA and infectious virus) in samples, and tissues as well as a reduction of the lung tissue damage assessed on day 7 post-exposure. Overall, the work is original and topical. Although the data are very promising, the conclusions lack statistical support.

Please note that we have provided statistical analyses to support essentially all of the measures in this manuscript, as detailed below.

Further, it is unfortunate that the data are confounded by the administration of concomitant medication to the nonhuman primates exhibiting clinical signs of disease. There are also shortcomings in the experimental design which could have been avoided.

We have also addressed this point below in the detailed responses.

Major comments

- It is my opinion that analgesic treatment with meloxicam (COX-2 inhibitor) should not have occurred. Concomitant medication must be avoided when evaluating the efficacy of a therapeutic and it is a serious experimental flaw that biases the data collected within the respective groups. If NHPs were sick enough to meet the euthanasia criterion, then they needed to be euthanized. Alleviating discomfort from clinical infection is not acceptable if this potentially compromises the integrity of the scientific data collected. I encourage the authors to discuss this feedback with institutional stakeholders on how to avoid this in their future studies (for example by adjusting the IACUC protocol and USDA pain and distress categories). As the authors state in lines 343-345, the goal was to observe clinical effects using nebulized mAb delivery following virus exposure; unfortunately, this goal must be considered confounded after treating the sickest NHPs with meloxicam on days 5 and 6, even if the data are still considered “in line” with what was generally expected/favored.

We want to clarify to the reviewers that this experiment was designed to be able to compare virus replication and pathology at matched timepoints in all animals so that we could compare treated animals with control animals. For this study, we chose to euthanize the animals at an experimental endpoint with necropsy and tissue collection at 7 days post-infection. This

endpoint was selected because we were comparing the results of this study with another study in which tissue was collected at 7 days post-infection, which had previously been shown to be a time where pathology could be distinguished in rhesus macaques. The animals were treated with an appropriate analgesic very late in the experiment, after the acute phase and not earlier than day 5, when they exhibited signs of discomfort, but none reached the humane endpoints approved by our IACUC; therefore, they did not require euthanasia. If relief of pain or distress were withheld, the protocol would have been classified as a USDA Category E study. Category E studies are not allowed in nonhuman primates at this institution. Alleviating discomfort from clinical infection is required for NHP studies, and we believe that there was no compromise of the scientific data. To avoid any bias, it is important that the criteria to initiate therapy with analgesics (or other drugs) are used consistently across all study arms. This is exactly what we did in this current study. It was the veterinary staff (and not the investigators) who made such treatment decisions, solely based on clinical signs irrespective of the treatment arm.

The treatment allowed the rigorous experimental design to be followed and was essential to alleviate discomfort. We did not administer corticosteroids, for example, which could have affected disease directly. NSAID of choice could have been Tylenol. It is critical to note that the administration of Meloxicam was done in one animal in the control group on days 5 and 6 and in two animals in the post-exposure experimental group on day 6 only, but the time of administration was after peak viremia and cytokine responses. We posit that damage to lungs was already well advanced by day 7. The post-exposure group was the least aided by the antibody treatment, which goes contrary to the thinking that Meloxicam improved clinical outcomes. Improved clinical outcomes were most pronounced in animals and groups that were not treated at any time with Meloxicam.

- Statistical analysis with n=4 NHPs does not provide sufficient power, and should be replaced with a description of findings instead. It is unfortunate that the authors cannot go back and re-design the study. My wish and suggestion would have been to focus on either the preventative or therapeutic aspect of aerosol mAb delivery (the one of higher importance), and combine the two control groups into one control arm only. Did the authors have a sufficiently high number of historical controls and resulting data on SARS-CoV-2 natural history (untreated)? The anti-RSV mAb control group is important for the author's pathology readout, but the NHPs in this group should have received the mAb not once, but with the highest frequency corresponding with the respective anti-SARS-CoV-2 cocktail. The total number of NHPs (n=20) could have been distributed across two groups total with n=10; one control group exposed to SARS-CoV-2 and receiving three doses of the anti-RSV control mAb (using the same frequency as the test group). The test group (for therapeutic evaluation) could have received the mAb cocktail once pre- and twice post-exposure. Alternatively, a n=7 (preventative), n=7 (therapeutic) and n=6 (control) grouping design would also have been possible and would have provided better power. The goal should have been to show high-powered efficacy data for the administration strategy that is of most relevance.

This experiment was not designed to be a definitive test of aerosol antibodies, as it is the first study to test anti-SARS-CoV-2 antibodies delivered by this route in NHPs. Prior studies had

shown that high doses of mAbs are protective if given prophylactically (intramuscularly) and ameliorate pathology in the lung when given therapeutically (e.g., Loo et al. ²). The lack of data on feasibility of aerosol delivery, coupled with our PK studies, indicated that only low doses would be delivered, and thus we wanted to test several scenarios to enhance antibody residence time. Prior to this study, SARS-CoV-2 delta strain had not been tested in rhesus macaques at our institute nor reported by others, so that we were not able to use historical controls from other centers using this same strain. The study design and statistical justification was peer-reviewed, and the number of animals was considered valid based on virological and pathological outcomes observed with other published SARS-CoV-2 NHP studies with the Washington strain, which we anticipated would be similar to the delta strain. Prior studies had shown that pre-exposure with antibodies delivered intravenously could affect disease outcome, using 4 animals per group. We chose four animals per group to be able to determine whether any efficacy could be observed delivering antibodies by aerosol, either as pre-exposure or as post-exposure. We have added a sentence in the Results section and in the Discussion to emphasize this point. As we are sure that the reviewer is aware, the success of antibodies as post-exposure therapy in the clinic has been modest, even when the antibodies are effective against the prevalent and infecting strain. It appears that timing is critical for antibodies to blunt infection. We learned that antibodies can be delivered immediately to the site of first infection, and that they can significantly blunt infection. We also learned that aerosol delivery is improved by multiple dosing, which is related to rapid clearance from lungs and nasal passages.

The study was designed to achieve statistical significance, and this was achieved in three different measures, despite the small sample sizes, as emphasized in the revised paper. This experiment was designed to measure differences in viral RNA and in infectious virus, both of which were achieved in BAL by combining the control groups. We also found statistically significant differences in cytokine measures. We would note that other studies have been published in very high-profile journals that included smaller numbers of animals per group (McMahan et al., Nature ³). We agree that there could have been a different distribution of numbers of animals per group, and future experiments can now be designed with our results in mind. The current study provides important information about antibody residence in the nasal passages and the lung, and it also provides surprising information that antibodies do not need to be delivered by the intravenous route to be effective in reducing viremia, inflammation, or pathology.

- Males and female animals needed to be balanced across groups. Why did the authors choose to assign males to Groups 1 and 2 and females to Groups 3 and 4 given that there were equal numbers of animals from both sexes?

Males and females were balanced in the control groups, but this occurred by chance. Due to space limitations and the logistics of handling infected animals under ABSL3 conditions, it was important to work so that animals could be phased in sequentially. We did not have the luxury of balancing the sexes due to extreme shortages of animals for this work. Appropriate animals were assigned as they became available. This is not an ideal situation, but it is a current reality

of the system, both at our institution and at other National Primate Research Centers.

Minor comments

- Consider rephrasing the paragraph on vaccines within the introduction, and/or attenuating the limitations of active immunization/vaccines. There are no real alternatives for vaccination as a preventative strategy to boost herd immunity, and it is the most accessible and cost effective way to achieve immunity within a population. The authors propose aerosol delivery of their mAb cocktail predominantly for therapeutic purposes and to lesser extent for a preventative one (or on a smaller scale to protect high-risk individuals), so this approach is in no real “competition” to vaccines. I suggest re-framing this context.

We agree that reframing is a good idea and have addressed this point in the Introduction. We can appreciate the point that vaccines are typically far more accessible and cost-effective as a means of prevention by inducing immunity. However, with SARS-CoV-2, people who are vaccinated can be reinfected, and many of them may do better with treatment, but we have removed this alternative to vaccine argument. In the Discussion, we speculate that for those high-risk individuals, there could be an advantage to take mAbs prophylactically as short-term protection if they were available and easy to administer. We understand that medical progress is not at this point.

- The result section on the histopathology data is lengthy (descriptive/qualitative histopathology findings) and could be shortened and be more focused.

As reviewer #1 pointed out, pathology is the main outcome of this treatment. We wanted to make this section as clear as possible so that reduction in disease can be clearly defined and understood by the readers. As noted by the other reviewer “...antibody treatment had profound effects in the pathology of the lung, infiltration of immune cells, outcomes that can be measured only in such an in depth and invasive study.” We think it is important to retain this level of detail to justify the conclusions and the title of the manuscript.

- Consider rephrasing general statements such as “there is encouraging news”, “modulating disease”, “right tissues” and “wreaks havoc with lung tissue” for example.

Thank you for these helpful suggestions, each of which has been addressed.

- Consider rephrasing COVID Post-exposure group to Virus post-exposure group. COVID-19 is reserved to describe the disease in humans.

Agreed, the reviewer is correct. We have corrected this point in the figures and legends and text.

References cited

1. Upadhyay, A.A., *et al.* TREM2(+) and interstitial-like macrophages orchestrate airway inflammation in SARS-CoV-2 infection in rhesus macaques. *Nat Commun* **14**, 1914 (2023).
2. Loo, Y.M., *et al.* The SARS-CoV-2 monoclonal antibody combination, AZD7442, is protective in nonhuman primates and has an extended half-life in humans. *Sci Transl Med* **14**, eabl8124 (2022).
3. McMahan, K., *et al.* Correlates of protection against SARS-CoV-2 in rhesus macaques. *Nature* **590**, 630-634 (2021).

Reviewers' Comments:

Reviewer #1:

Remarks to the Author:

In my view the authors have adequately addressed the issues I consider important and improved the manuscript. It should be ready for publication.

Reviewer #3:

Remarks to the Author:

The manuscript by Streblow et al is very well written and highly responsive to prior critiques. The authors thoroughly explain in detail the difficulties that are associated with conducting such challenging studies in NHPs. Statistical power and the ability to distribute equal numbers of animals of both genders in different experimental groups is limited in NHPs, as is the ability to derive different types of answers (survival vs time-matched comparison of pre-specified assays which are the major discriminant in the minds of the authors). Despite these limitations, this group of authors did their best to attempt to generate valuable information.

Their results clearly show that passive treatment with aerosolized monoclonal antibodies (pre or post exposure) with previously determined high levels of neutralizing activity against SARS-CoV-2 results in:

1. Reduced bacterial loads in the lung compartment, particularly in lower respiratory tract (BAL, black and green groups, Fig 2A-D, left column).
2. Reduced spread of established viral infection to other regions of the lung compartment (Fig 3A).
3. Reduced lung pathology (Fig 4D), and
4. Reduced levels of several molecular markers of inflammation and lung damage (Fig 5).

These results are well discussed and have implications for the continuing treatment of the COVID-19 pandemic, as well as for future pandemics that may emerge. These results will go a long way in establishing clinical evaluation of this treatment.

These results clearly indicate that there is high degree of value in

REVIEWERS' COMMENTS

Reviewer #1 (Remarks to the Author):

In my view the authors have adequately addressed the issues I consider important and improved the manuscript. It should be ready for publication.

Author response: thank you for the review and we agree that the manuscript is improved following earlier recommendations.

Reviewer #3 (Remarks to the Author):

The manuscript by Streblow et al is very well written and highly responsive to prior critiques. The authors thoroughly explain in detail the difficulties that are associated with conducting such challenging studies in NHPs. Statistical power and the ability to distribute equal numbers of animals of both genders in different experimental groups is limited in NHPs, as is the ability to derive different types of answers (survival vs time-matched comparison of pre-specified assays which are the major discriminant in the minds of the authors). Despite these limitations, this group of authors did their best to attempt to generate valuable information.

Their results clearly show that passive treatment with aerosolized monoclonal antibodies (pre or post exposure) with previously determined high levels of neutralizing activity against SARS-CoV-2 results in:

1. Reduced bacterial loads in the lung compartment, particularly in lower respiratory tract (BAL, black and green groups, Fig 2A-D, left column).
2. Reduced spread of established viral infection to other regions of the lung compartment (Fig 3A).
3. Reduced lung pathology (Fig 4D), and
4. Reduced levels of several molecular markers of inflammation and lung damage (Fig 5).

These results are well discussed and have implications for the continuing treatment of the COVID-19 pandemic, as well as for future pandemics that may emerge. These results will go a long way in establishing clinical evaluation of this treatment.

These results clearly indicate that there is high degree of value in

Author response: We appreciate the favorable review and the understanding of the reviewer, given the difficulties of NHP studies in ABSL-3 conditions. Thank you for the comments.